# A 414-year tree-ring based April-July minimum temperature reconstruction and its implications on the extreme climate events, northeast China

Shanna Lyu[1], Zongshan Li[2], Yuandong Zhang[3], Xiaochun Wang[1, *]

[1] Center for Ecological Research, Northeast Forestry University, Harbin 150040, China

[2] State Key Laboratory of Urban and Regional Ecology, Research Center for Eco-Environmental Science, Chinese Academy of Sciences, Beijing 100085, China

[3] Key Lab of Forest Ecology and Environment, State Forestry Administration, Institute of Forest Ecology, Environment and Protection, Chinese Academy of Forestry, Beijing 100091, China

*Correspondence to*: Xiaochun Wang, Center for Ecological Research, Northeast Forestry University, Harbin 150040, China. E-mail: wangxc-cf@nefu.edu.cn.

**Abstract:** A ring-width series was used as a proxy to reconstruct the past 414-year record of April-July minimum temperature at Laobai Mountain, northeast China. Chronology was built using standard tree-ring procedures for providing comparable information in this area while preserving low-frequency signals. By analyzing the relationship between the tree-ring chronology of Korean pine (*Pinus koraiensis*) and meteorological data, we found that the standard chronology was significantly correlated with the April-July minimum temperature ($r = 0.757$, $p < 0.01$). Therefore, the April-July minimum temperature since 1600 (more than six trees, but EPS is greater than 0.85 since 1660) was reconstructed by this tree-ring series. The reconstruction equation accounted for 57.3% of

temperature variation, and it was proved reliable by testing with several methods (e.g., sign test, product mean test, reduction of error, and coefficient of efficiency). Reconstructed April-July minimum temperature in Laobai Mountain showed six major cold periods (1605-1616, 1645-1677, 1684-1691, 1911-1924, 1930-1942 and 1951-1969) and seven major warm periods (1767-1785, 1787-1793, 1795-1807, 1819-1826, 1838-1848, 1856-1873 and 1991-2008) during the past 414 years. The reconstructed low temperature periods in the 17[th] and early 18[th] century were consistent with the Little Ice Age (LIA) in the Northern Hemisphere, and the rate of warming in the 19[th] century was significantly slower than that in late 20[th] century. In addition, the reconstructed series was fairly consistent with the historical and natural disaster records of extreme climate events (e.g., cold damage and frost disaster) in this area. It exhibited 128-60-, 9.9-9.7-, 4.0-, 3.8-3.7-, 3.0-2.6-, 2.4- and 2.2- year periods of warm-cold changes. This temperature record provides a new evidence of past climate variability, and can be used to predict the climate trend in the future in northeast China.

**Keywords**   Minimum temperature reconstruction; tree-ring width; northeast China; *Pinus koraiensis*; extreme climate

# 1 Introduction

Global climate change presents major challenges for humans and the natural systems that provide ecosystem services. Consequently, it is urgent to better understand climate change and its forcing mechanisms. Instrumental records are typically less than 100 years and often less than 50 years in most areas of the world. It is necessary to put the present climate regime in context with the long-term perspectives, which forces a reliance on natural proxy records to reconstruct the past climate. Tree rings have been widely applied in global climate change studies and paleoclimate reconstructions at both regional and global scales because they offer accurate and continuity temporal record as well as they are widespread and easily replicated (Corona et al., 2010; Bouriaud et al., 2014; Kress et al.,

2014).

Northeast China, an area sensitive to global climate change, is located in the ecotone from a temperate to cold temperate zone, belonging to monsoon fringe area. Due to the interannual instability of monsoon, frequent climate extremes (especially cold damage or frost disaster) seriously affect agriculture and forest ecosystems. In addition, previous studies suggest that climate change in northeast China was also linked to the solar activities and global land-sea atmospheric circulation during certain pre-instrumental periods (Chen et al., 2006; Wang et al., 2011; Liu et al., 2013). It is generally accepted that the climate warms during periods of strong solar activity (e.g., Medieval Warm Period) and colds during periods of low solar activity (e.g., Little Ice Age) (Lean and Rind, 1999; Bond et al., 2001). Recently, the warming in northeast China has been significantly affected by the global warming since the 20[th] century (Ding and Dai, 1994; Wang et al., 2004; Zhao et al., 2009), which is often caused by a faster rise of night or minimum temperature (Karl et al., 1993; Ren et al., 1998; Tang et al., 2005). To explore whether the climate warming is abnormal and predict the future trend of temperature change in this area, we must fully understand the history of climate changes over a long period. However, tree-ring series were rarely used to reconstruct past climate (especially temperature) in this area because of the exceptional hydrothermal conditions. Several temperature-sensitive tree-ring chronologies were developed in Changbai Mountain (e.g., Shao and Wu, 1997; Zhu et al., 2009; Wang et al., 2012; Li and Wang, 2013) and Xiaoxing'an Mountain (Yin et al., 2009; Zhu et al., 2015), but almost no records were obtained for the period of over 250 years, which can reflect the low-frequency climate variations. This limits our understanding a longer time scale of climate regime in northeast China. Temperature reconstructions are also far from adequate and do not satisfy the demands of scientific research. Therefore, there is a requirement for higher-quality climate reconstructions in a greater number of areas over longer periods and a larger group of climatic indicators for verification in this region. For this reason, more information of regional past climate variations registered in a long-term tree-ring series is needed, and it is important to understand the impacts of climate change on forest ecosystems and the ecosystems services provided to humans in northeast

China.

Currently, a significant climate warming (especially the minimum temperatures increase) occurred in northeast China since the 1980s. However, there still remains a lack of long-term climatic record (at least more than 250 years) in this area to explore what is the temperature regime in the past one or half thousand years, and if the current warming is unprecedented. Therefore, the main objectives of this study are (1) to develop for the first time a more than 400-year ring-width chronology in northeast China; (2) to analyze the regime of temperature variation during the past four centuries in northeast China; (3) to identify the recent warming amplitude in a long-term context; (4) and to analyze the extreme low temperature events. Our new minimum temperature record supplements existing data in northeast China and provides a new evidence of past climate variability. There is the potential to better understand future climatic trajectories from these data in northeast China.

## 2 Materials and methods

### 2.1 Study area

The study area is located at Laobai Mountain (128°03' E, 44°06' N), the boundary zone between Jilin and Heilongjiang provinces, and is also an ecotone between Changbai and Xiaoxing'an Mountain. Laobai Mt. is the third highest peak in northeast China and rises to 1650 m above sea level (a.s.l.). Almost no inhabitants live in or near the Mountain, so the forest ecosystem is preserved very well and the native vegetation remains predominantly intact (Fig. 1). Five forest vegetation types from temperate to frigid change with the altitude increase, which is the broad-leaved *Quercus mongolica* forest below 800 m a.s.l., the mixed broadleaved Korean pine forest from 800 to 1050 m, dark conifer forest with *Picea jezoensis* from 1050 to 1350 m, *Betula ermanii* forest between 1350 and 1640 m, and *Pinus pumila* forest and subalpine meadow above 1640 m. Plant flora is a transition from Changbai Mountain to Xiaoxing'an Mountain. Five tree species were cored in this area, but only Korean pine (*Pinus koraiensis*) cores were used in this study. Korean pine is a sun-loving plant (shade tolerant when it is young) and

has shallow roots, widely distribute on well-drained wet mountain slopes close to the subalpine timberline where the brown forest soil is covered. The forest vegetation in sampling area is the mixed broadleaved Korean pine forest dominated by *Pinus koraiensis*, *Picea jezoensis* and *Abies nephrolepis* as well as broadleaf tree species, such as *Juglans mandshurica*, *Fraxinus mandshurica* and *Acer mono* (Bu et al., 2003).

This region belongs to temperate continental monsoon climate. Climate data are collected from the nearest meteorological station in Dunhua. The mean annual temperature from 1956 to 2013 is 3.3 °C, with July (20.1 °C) and January (-16.8 °C) being the warmest and the coldest month, respectively. The mean monthly minimum and maximum temperatures are -2.5 and 9.8 °C, respectively. The mean annual total precipitation is 627 mm, the majority (63.1%) of which falls during June-August. The annual frost-free period is approximately 90-110 days (Fig.

2).

## 2.2 Tree-ring chronology development

Korean pine tree-ring samples were obtained from the south slope of Laobai Mountain along an elevational gradient from 950 to 1050 m, from an almost pristine area containing well-preserved old forests largely uninfluenced by human activity. One or two cores per undamaged tree (71 cores from 41 trees) were extracted from cross-slope sides

of the trunks at breast height using an increment borer. Cores were air dried, glued firmly to groove wooden mounts and sanded with progressively finer grade abrasive paper up to 800 grit. Then, the samples were cross-dated using a skeleton plot method (Stokes and Smiley, 1968), each tree-ring width was measured with a precision of 0.001 mm using VELMEX tree-ring width measurement system (Velmex, Inc., Bloomfield, NY, USA). Data were checked for missing or false rings and dating errors using the quality control program COFECHA (Holmes, 1983).

ARSTAN program was used to detrend and standardize cross-dated tree-ring width series into a tree-ring chronology (Cook, 1985). During this detrending process, to remove biological factors (such as age-related trends) and non-climatic variations and preserve as much low-frequency signal as possible, each ring-width series were

fitted with a straight line or negative exponential function. A 67% cubic smoothing spline with a 50% cutoff frequency was further used in a few cases when anomalous growth trends occurred. The detrended data from individual tree cores were then averaged using a bi-weight robust mean to develop the standard (STD), residual (RES) and autoregressive (ARS) chronologies (Cook and Kairiukstis, 1990).

Statistical characteristics for the STD and RES chronologies of *Pinus koraiensis* in Laobai Mountain is shown in Table 1. These statistic characters of tree-ring chronologies contained strong climate signals, common growth-limiting signals and the amount of different frequency information. As shown in Fig. 3, the amplitude of STD chronology (Fig. 3a) in low-frequency variability was larger than that in RES chronology (Fig. 3c). This indicated that STD chronology preserved more low-frequency signals, while RES chronology reflected high-frequency signals. The mean sensitivity of RES chronology was larger than STD chronology, which quantitatively illustrated that RES chronology exhibited more high-frequency climate information than STD chronology did (Table 1). The full length of tree-ring series spanned from 1600 to 2015. The expressed population signal (EPS) was used to assess the quality of STD chronology (Wigley et al., 1984). A generally acceptable threshold of the EPS was consistently greater than 0.85 from AD 1660 to 2015 (eleven trees) (Fig. 3b), which affirmed that this is a reliable period. However, although the EPS value from AD 1600 to 1659 was less than 0.85, it matches a minimum sample depth of six trees in this segment. It is very important to extend the reconstruction tree-ring chronology as possible as we could because of few long climate reconstructions in this area. Therefore, we kept this part from 1600 to 1659 to be used in the reconstruction. In addition, the STD chronology was used in the subsequent analyses to obtain more low-frequency signals.

**2.3 Climate data and statistical methods**

Meteorological data were obtained from the National Meteorological Information Center (http://data.cma.cn/). Considering the proximity to sampling sites and climate record length, climate data from Dunhua meteorological

station (43°22′ N, 128°12′ E, elevation 524.9 m a.s.l., 1956-2013) were selected to identify climate signals in the tree-ring series. The climate variable included monthly total precipitation, mean maximum temperature ($T_{max}$), mean temperature ($T_{mean}$), and mean minimum temperature ($T_{min}$). Months from the previous July to current August were selected for the analysis of the relationship between climate variables and Korean pine growth.

To identify climate-growth relationships of Korean pine in Laobai Mountain, a Pearson's correlation was performed between climate variables and tree growth. The stability and reliability of the reconstruction equation was assessed by the split-period calibration and verification analyses (Fritts, 1976; Cook and Kairiukstis, 1990) for the two periods 1956-1984 and 1985-2013. The Pearson's correlation coefficient ($r$), $R$ square ($R^2$), sign test ($ST$), the reduction of error ($RE$), coefficient of efficiency ($CE$) and product means test ($PMT$) are the tools used to verify the

results. To test the periodicity of the reconstructed minimum temperature in this area, a spectral analysis was performed using a multi-taper method, especially powerful for time series (Mann and Lees, 1996). All statistical analyses presented in this paper were performed using a commercial software SPSS12.0 (SPSS, Inc., Chicago, IL, USA).

**3 Results and discussion**

**3.1 Climate-radial growth relationship**

Relationships between the STD and RES chronologies and monthly climate data in Dunhua were shown in Fig. 4. Temperatures were more crucial to Korean pine growth compared with precipitation. In contrast, the correlation coefficients between Korean pine chronologies and mean minimum temperature were positive and higher than those for maximum and mean temperature. The significant correlation months between STD chronology (Fig. 4a) and

mean minimum temperature were not found in the RES chronology (Fig. 4b). This indicated that the STD chronology recorded the minimum temperature signals in low frequency, but not at high frequencies. Different

combinations of months were also considered (Table 2). The best-correlated three-month season, April-July ($r$ = 0.757, $p$ < 0.0001), was then selected for temperature reconstruction of the mean minimum temperature (MMT) (Table 2).

It was generally accepted that extreme temperatures limited tree growth at treeline or at high latitudes forest, especially spring or early summer minimum temperature (Wilson and Luckman, 2002; Körner and Paulsen, 2004; Porter et al., 2013; Yin et at., 2015). Moreover, $T_{max}$, $T_{mean}$ and $T_{min}$ during the observed period of 1956-2013 illustrated similar inter-annual variations (Fig. 5), while the increase trend of $T_{min}$ was much higher than $T_{mean}$ and $T_{max}$, especially since 1976. This phenomenon was consistent with the results in Karl et al. (1993), Ren et al. (1998) and Tang et al. (2005). They indicated that climate warming over past decades was mostly owing to the faster rise of night or minimum temperatures. This seemed to be the case in northeast China as well. Based on the relationship between STD chronology and climate data, we found that the minimum April-July temperature played more important roles in limiting Korean pine radial growth in Laobai Mountain compared with the maximum and mean temperatures. This also meant that less warm and wet conditions in this area were suitable for Korean pine growth.

This may arise from two reasons. One, the sampling site was located at high elevations close to the upper limit of Korean pine distribution, which may cause tree growth more sensitive to minimum temperature (Szeicz and MacDonald, 1995; D'Arrigo et al., 2009; Li et al., 2011; Yu et al., 2011; Flower and Smith, 2012). High minimum temperatures in early growing season can inhibit frost damage, and thus allow the formation of a wider ring (Wu, 1990; Akkemik, 2000; Mäkinen et al., 2003). High nighttime temperatures can also promote tree respiration and enhance physiological activities, thereby producing more auxin, promoting cell enlargement, and forming a wider ring in growing season (Fritts et al., 1976). Increasing temperature may allow trees to conduct photosynthesis at the early stage of the growing season, which might produce more auxin production. Another, a crucial growth period of Korean pine in every year was from April to July. During this period, temperature could have direct effects on photosynthesis rate, cambium activity, and respiration efficiency, etc., which affecting the formation of ring width

(Li et al., 2000; Yu et al., 2011). Therefore, Korean pine radial growth was positively correlated with the average minimum temperature from April to July.

## 3.2 Minimum temperature reconstruction

Based on above analysis, a linear regression equation was established to reconstruct the April-July MMT. The transfer function was as follows:

$$Y = 2.987X_t + 4.829$$

($N = 58$, $R = 0.757$, $R^2 = 0.573$, $R^2_{adj} = 0.565$, $F = 75.161$, $p < 0.0001$)

where $Y$ was the April-July MMT and $X$ was the tree-ring index of Korean pine chronology at year $t$. As shown in Fig. 6a, the reconstructed values closely tracked the observed temperature. The calibration and verification statistics were shown in Table 3. Parameters of the split-sample validation periods indicated the reconstruction equation was stable in the whole period. Positive RE and CE values revealed a useful paleoclimatic information (Cook et al., 1999). Significant results of ST and PMT indicated a good agreement between the actual and reconstructed data. However, the first difference correlation (not shown) between the STD chronology and temperature did not exceed the 95% confidence level. This confirmed that the STD chronology might better capture the low-frequency variability rather than high-frequency variability. In addition, the correlation coefficient between the first-order difference series of the actual and reconstructed values were not significant at the 0.05 level ($r = 0.12$, $p > 0.05$), hence, this reconstructed minimum temperature series was more consistent with the observed series at low-frequency variability. The final calibration equation accounted for 57.3% of the total variance ($p < 0.0001$) and passed all calibration and verification statistical requirements. Hence, this equation was reliable and allowed for the accurate reconstruction of the April-July MMT in Laobai Mountain.

## 3.3 Temperature variations from 1600 to 2013 AD

The reconstructed average April-July MMT variations since 1600 AD and its 11-year moving average was shown in Fig. 6b. The 11-year moving average of the reconstructed series was used to obtain low-frequency information and analyze temperature variability in this region. Mean value of the 414-year reconstructed temperature was 7.66 °C, with a standard deviation of ± 0.53 °C. The warm and cold periods were defined when temperatures exceeded the mean value plus and minus 0.5 times standard deviation, respectively (Fig. 6b). The reconstructed April-July MMT series exhibited six cold and seven warm periods. The longest cold period lasted from 1645 to 1677 AD (33 years), with an average temperature of 0.5 °C below the mean value. The longest warm period, however, lasted from 1767 to 1785 AD (19 years), and the average temperature was 0.69 °C above the mean value (Table 4). Four cold (1605-1616, 1645-1677, 1911-1924, and 1951-1969) and warm (1795-1807, 1838-1848, 1856-1873 and 1991-2008) periods were consistent with other results of tree-ring reconstructions in northeast China (Shao and Wu, 1997; Yin et al., 2009; Wang et al., 2012; Zhu et al., 2015). In addition, two cold periods 1645-1677 and 1684-1691 were consistent with the Maunder Minimum (1645-1715), an interval of decreased solar irradiance (Bard et al., 2000). The cold period 1645-1677 also appeared in other proxy records of reconstructed temperatures, which coincided with the Little Ice Age (LIA) in the northern hemisphere (Lin et al., 2004; Wang et al., 2006; Hong et al., 2009). Cold conditions during the 17th century and the rapid warming during the mid-19th and late 20th century in northeast China was present in this series, suggesting it could be a good proxy for regional temperature variations in northeast China.

To further evaluate the reliability of this reconstruction, we compared our reconstruction series with two nearby tree-ring based reconstruction temperature series from Dunhua (Li and Wang, 2013; Fig. 7a) and Changbai Mountain (Zhu et al., 2009; Fig. 7b) and the northern hemisphere temperature reconstruction (D'Arrigo et al., 2006; Fig. 7d) (Fig. 7). Interestingly, a significant negative correlation ($r = -0.18$, $p < 0.01$) between our reconstruction (Fig. 7c) and the northern hemisphere temperature reconstruction (D'Arrigo et al., 2006) was found (Fig. 7d), while our reconstruction of April-July MMT had the similar variations in the April-September temperature reconstruction

in Dunhua ($r = 0.50$, $p < 0.01$; Fig. 7a) and the February-April temperature reconstruction in Changbai Mountain ($r = 0.45$, $p < 0.01$; Fig. 7b). The three temperature series exhibited significantly low temperature periods during the 1950s-1970s, which coincided with a slight decrease in solar activity from 1940-1970 AD (Beer et al., 2000) (Fig. 7).

It was widely believed that the LIA in China exhibited three cold periods in the 15th, 17th, and 19th centuries (Wang et al., 2003), and this was confirmed by our reconstruction series (Fig. 7c and Table 4). The first cold period in our series was less obvious, while the second one was the most obvious of all. A different beginning and ending year of the second cold period in our reconstruction was found (Fig. 7c and Table 4). In addition, there existed a regional difference for the third cold period, that is, it was obvious in south China, while had opposite phase in northeast

China (Wang et al., 1998; Wang et al., 2003). The third cold period in 19th century was not obvious in our reconstruction, which was consistent with Wu (2013) and Wang et al. (1998). This also leaded to a bad matching with the Northern Hemisphere temperature (D'Arrigo et al., 2006). Another notable feature in Fig. 7 was a sharp temperature increase since the 1980s, and rose to the peak in the early 2000s. Temperature increase in this area was consistent with the report from the Intergovernmental Panel on Climate Change (IPCC, 2007). These series

displayed similar patterns of low-frequency variations suggesting that the reconstructed temperature in northeast China was significantly correlated with large-scale variations (Fig. 7). The reconstructed April-July MMT in the past 414 years showed 128-60-, 9.9-9.7-, 4.0-, 3.8-3.7-, 3.0-2.6-, 2.4- and 2.2-year quasi-cycles at a 95% confidence level (Fig. 8).

Unfortunately, three compared temperature series also showed dissimilar variations in some cold/warm years (Fig. 7). This might be due to differences in reconstructed temperature months, parameters (such as mean, minimum, and maximum temperature), and habitat conditions in different sampling areas. Recent studies suggested that the mean, minimum and maximum temperature variations were often asymmetric (Karl et al., 1993; Xie and Cao, 1996;

Wilson and Luckman, 2002; Wilson and Luckman, 2003; Gou et al., 2008). Global warming over the past decades was mostly owing to the faster rise of night or minimum temperatures, but not maximum temperature. The unsynchronized variability among mean, minimum and maximum temperatures was found in Dunhua meteorological station (Fig. 5). The sampled site was located at a junction zone between Jilin and Heilongjiang provinces, further north than Changbai Mountain. Meanwhile, some differences of the reconstructed temperature series were explained reasonably well from the comparison with analogous regions. Consequently, these findings could reveal more characteristics of regional climate variations and provide reliable data for larger-scale climate reconstructions in northeast China.

**3.4 Detection of northeast-wide cold damage or frost disaster events**

As the minimum temperature approached or fell below freezing point, it may limit biological activity and growth. Therefore, years with low temperatures were often accompanied by cold damage or severe frost events. The evidence from historical documents (Sun et al., 2007) showed that cold damage or frost disaster events occurred in Heilongjiang Province since 1675 (Table 5). Extremely cold damage or frost disaster events were in good agreement with eight low temperature years (1675, 1682, 1689, 1699, 1730, 1748, 1812, and 1885 in Fig. 6b and Table 4) in reconstructed April-July MMT series during the 1600s-1800s. At the beginning of 20[th] century, three severe frost periods occurred in the periods 1909-1918, 1934-1945, and 1956-1972 in Heilongjiang Province (Sun et al., 2007) were represented in our reconstruction (Fig. 6b and Table 4). In addition, other low temperature years in our reconstruction corresponded to extreme frost disaster events occurred in the periods 1902-1903, 1912-1914, 1920, 1932, 1934-1936, 1940, 1947, 1956-1961, 1964-1965, 1967, and 1969 (Fig. 6b and Table 4). The results revealed that 27 of the 30 cold damage or frost disaster events corresponded to the April-July MMT lower than the 27-year moving average, while the remaining three events corresponded to higher than April-July MMT values. In contrast, we found a decreasing trend of the annual extreme low temperature frequency and cold damage or severe

frost events with the warming since the 1980s. In summary, the reconstructed April-July MMT in Laobai Mountain strongly revealed the cold damage or frost disaster events in the past 414 years.

## 4 Conclusions

A significant positive correlation between the tree-ring width of Korean pine and the April-July MMT was found in Laobai Mountain, northeast China, and the April-July MMT was reconstructed for the past 414 years (1600-2013). The reconstructed and instrumental temperature series exhibited coherence over the common periods. The reconstructed series showed interannual to multidecadal temperature variations over the past 414 years. The cold/warm periods of the reconstructed minimum temperature record were also observed in historical documents and several proxy temperature records in northeast China. The most notable feature of the reconstructed series was obviously rapid warming trend since the 1980s, which was also confirmed by other reconstructed temperature series. Additionally, the correspondence between the low-temperature years and the historical cold damage or severe frost events demonstrated the potential relationship between April-July MMT and extreme cold events. This temperature may provide new and valuable information for the longest temperature variations period in northeast China.

**Acknowledgments** This research was supported by the National Natural Science Foundation of China (Nos. 41471168 and 31370463), the Key Project of the Special Focus on "Global Change and Response" of the China National Key Research and Development Plan (2016YFA0600800), the Program for Changjiang Scholars and Innovative Research Team in University (IRT-15R09), and the Program for New Century Excellent Talents in University (NCET-12-0810). We greatly appreciate the three anonymous referees for their constructive and helpful comments in revising and improving our manuscript a lot. We thank the staff of Laobai Mountain Forestry Bureaus for their assistance in the field. Meanwhile, we greatly appreciate Neil Pederson at Harvard Forest, Harvard

University for his assistance in editing parts of the English language.

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

**Tables**

Table 1. Major statistical characteristics for the chronology of *Pinus koraiensis* in Laobai Mountain.

|  | STD | RES |
|---|---|---|
| Number of cores | 71 | 71 |
| Time span | 1600-2015 | 1600-2015 |
| Mean sensitivity | 0.12 | 0.14 |
| Standard deviation | 0.20 | 0.12 |
| Correlations between trees | 0.22 | 0.28 |
| Correlations within trees | 0.68 | 0.51 |
| Signal-to-noise ratio | 8.72 | 11.81 |
| Autocorrelation order1 | 0.72 | 0.03 |
| Agreement with population chronology | 0.90 | 0.92 |
| Variance in first eigenvector | 28.51% | 30.03% |
| First year where EPS>0.85 (No. of trees) | 1660 (11) | 1685 (15) |

Table 2. Correlation coefficients between the STD chronology and the climate data of different month combinations during the common period of 1956–2013.

| Months | $T_{mean}$ | $T_{min}$ | $T_{max}$ |
|--------|-----------|-----------|-----------|
| c4-c7 | 0.577** | 0.757** | 0.177 |
| c4-c8 | 0.557** | 0.717** | 0.183 |
| c4-c9 | 0.599** | 0.726** | 0.217 |
| c5-c7 | 0.556** | 0.749** | 0.198 |
| c5-c8 | 0.522** | 0.691** | 0.198 |
| c5-c9 | 0.587** | 0.709** | 0.236 |
| c6-c8 | 0.447** | 0.634** | 0.199 |
| c6-c9 | 0.535** | 0.671** | 0.241 |
| p7-c8 | 0.586** | 0.682** | 0.230 |

* Significant at the 0.05 level (two-tailed). ** Significant at the 0.01 level (two-tailed).

Table 3. Calibration and verification statistics of the reconstruction equation for the common period of 1956-2013

| Calibration | $R$ | $R^2$ | Verification | $R$ | Reduction of Error | Coefficient of efficiency | Sign Test | Product Mean Test |
|---|---|---|---|---|---|---|---|---|
| whole Section (1956-2013) | 0.757** | 0.573** | - | - | - | - | - | - |
| Front Section (1956-1984) | 0.414* | 0.171* | Back Section (1985-2013) | 0.632** | 0.738** | 0.446** | (20, 9)* | 4.586** |
| Back Section (1985-2013) | 0.632** | 0.400** | Front Section (1956-1984) | 0.414* | 0.738** | 0.634** | (22, 7)** | 6.099** |

* Significant at the 0.05 level (two-tailed). ** Significant at the 0.01 level (two-tailed).

Table 4. Cold and warm periods based on the 11-year moving average April-July mean minimum temperature in Laobai Mountain during 1600-2013 AD.

| Rank | Cold period | | | Warm period | | |
|---|---|---|---|---|---|---|
| | period | Year | Mean (°C) | period | Year | Mean (°C) |
| 1 | 1605–1616 | 12 | 7.41 | 1767-1785 | 19 | 8.35 |
| 2 | 1645–1677 | 33 | 7.19 | 1787-1793 | 7 | 8.01 |
| 3 | 1684–1691 | 8 | 7.19 | 1795-1807 | 13 | 8.00 |
| 4 | 1911–1924 | 14 | 7.09 | 1819-1826 | 8 | 8.07 |
| 5 | 1930–1942 | 13 | 7.27 | 1838-1848 | 11 | 8.13 |
| 6 | 1951–1969 | 19 | 7.08 | 1856-1873 | 18 | 8.13 |
| 7 | | | | 1991-2008 | 18 | 8.38 |

Table 5. Cold damage or frost disaster events recorded in historical archives in Heilongjiang Province since 1675 (Sun et al., 2007).

| 17th century | 18th century | 19th century | 20th century |
|---|---|---|---|
| 1675 | 1730 | 1800-1801 | 1901-1903 |
| 1682 | 1746 | 1812-1813 | 1909-1915 |
| 1689 | 1749 | 1830-1832 | 1917 |
| 1699 | 1748 | 1878-1879 | 1920 |
| | 1755 | 1885 | 1925-1926 |
| | 1757 | | 1931-1932 |
| | | | 1934-1936 |
| | | | 1939-1943 |
| | | | 1947 |
| | | | 1950-1969 |
| | | | 1998-1999 |
| | | | 1971-1981 |

**Figure captions**

**Fig. 1** Map of the sampling site, the compared temperature series, nearly temperature series and meteorological station in northeast China. The photo showed the sampled site in Laobai Mountain and the remarkable vertical vegetation distribution along altitude changes.

**Fig. 2** Mean monthly temperature (°C) and total precipitation (mm) at Dunhua meteorological station for the period from 1956 to 2013.

**Fig. 3** Variations of the STD (a) and RES (c) chronology and sample depth, and the expressed population signal (EPS) and average correlation between all series (Rbar) of the STD (b) and RES (d) chronology from 1600 to 2015 AD.

**Fig. 4** Correlations between the monthly mean meteorological data (including mean temperature, mean maximum temperature, mean minimum temperature, and total precipitation) from Dunhua meteorological station (1956-2013) and (a) the STD chronology and (b) RES chronology, respectively. The dashed horizontal line represents the 95% confidence limit.

**Fig. 5** Inter-annual variation of the mean maximum (a), mean (b) and mean minimum temperatures (c) from 1956 to 2013. The straight line represents the fitted trend line, respectively.

**Fig. 6** (a) Actual (black line) and reconstructed (blue line) April-July minimum temperature for the common period of 1956-2013; (b) Reconstruction of April-July minimum temperature in Laobai Mountain for the last 414 years. The smoothed line indicates the 11-year moving average, and blue dots represent minimum freezing events.

**Fig. 7** (a) April-September mean minimum temperature reconstructed by Li and Wang (2013) in Dunhua, (b) February-April temperature established by Zhu et al. (2009) in Changbai Mountain, (c) April-July minimum temperature in Laobai Mountain, and (d) Northern Hemisphere extratropical temperature (D'Arrigo et al., 2006). Black lines denote temperature reconstruction values, and red color lines indicate the 11-year moving average.

**Fig. 8** Multi-taper method power spectra of the reconstructed minimum temperature (1600-2013 AD). Peaks above the dotted line represent significant periods at the $p < 5\%$.

**Fig. 1**

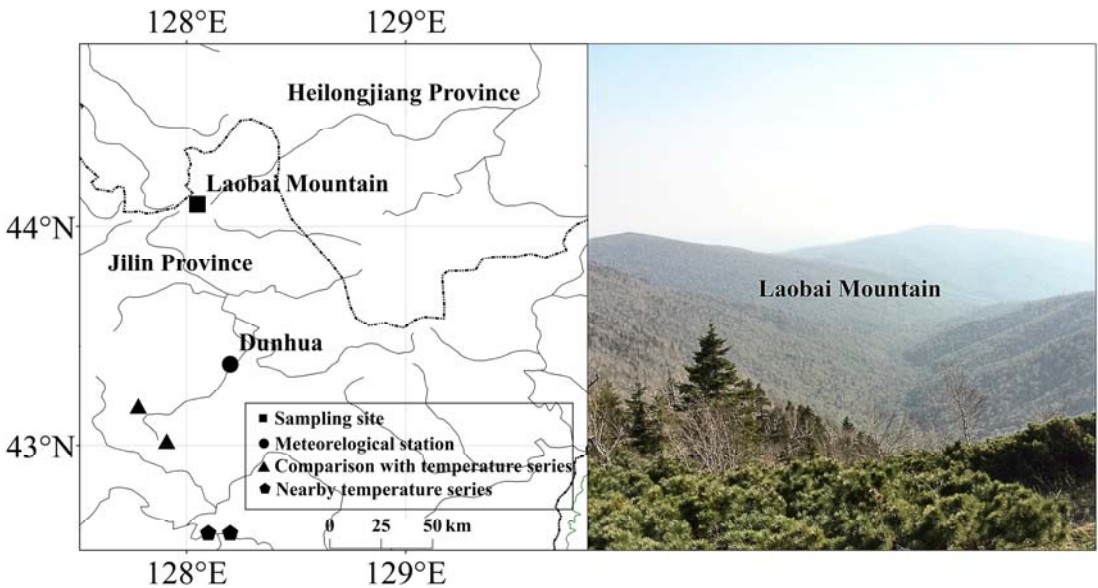

**Fig. 2**

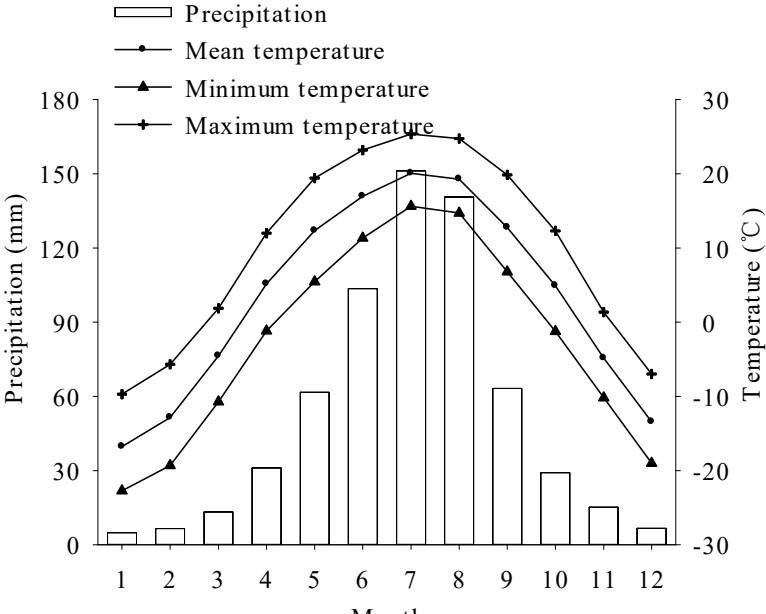

**Fig. 3**

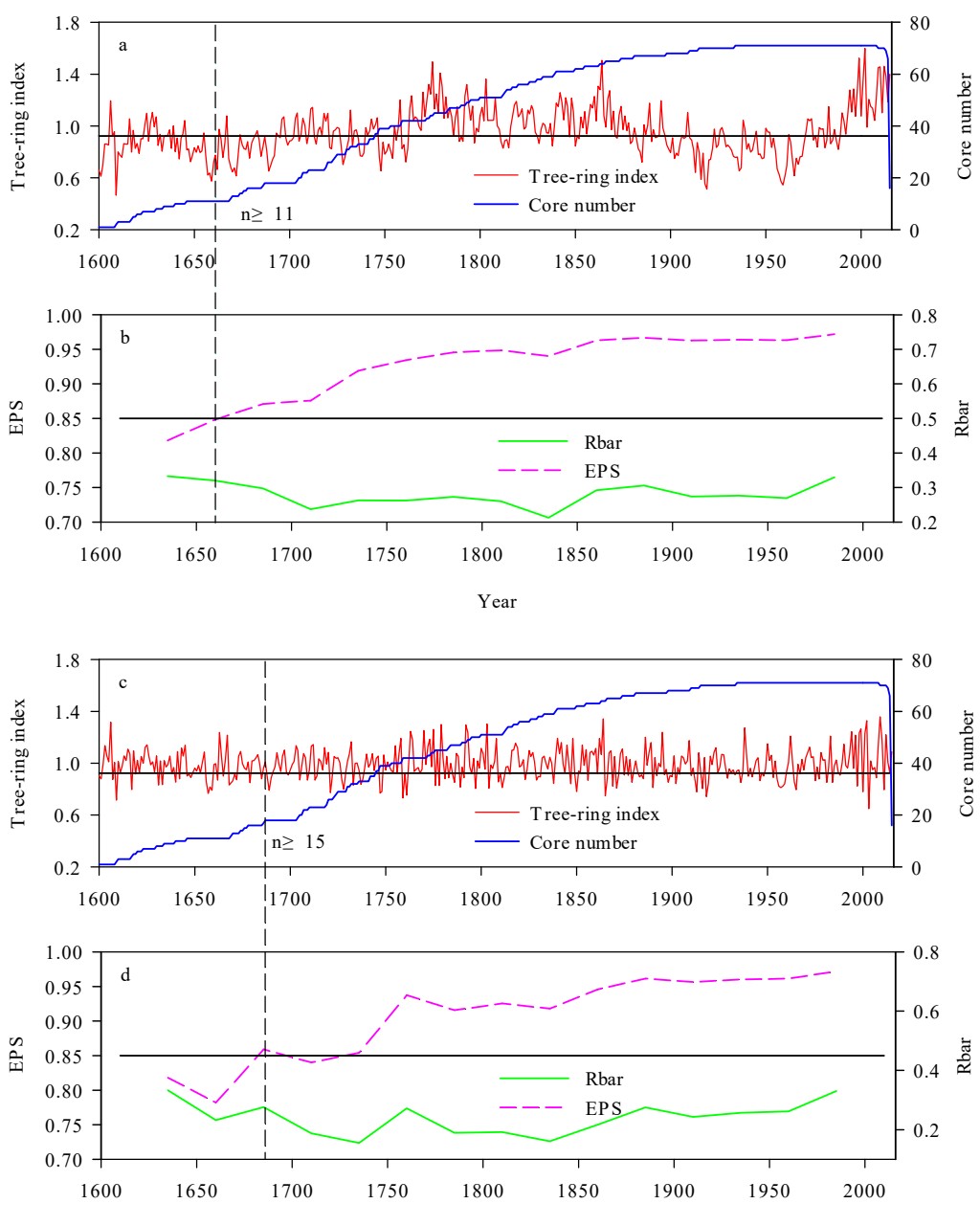

**Fig. 4**

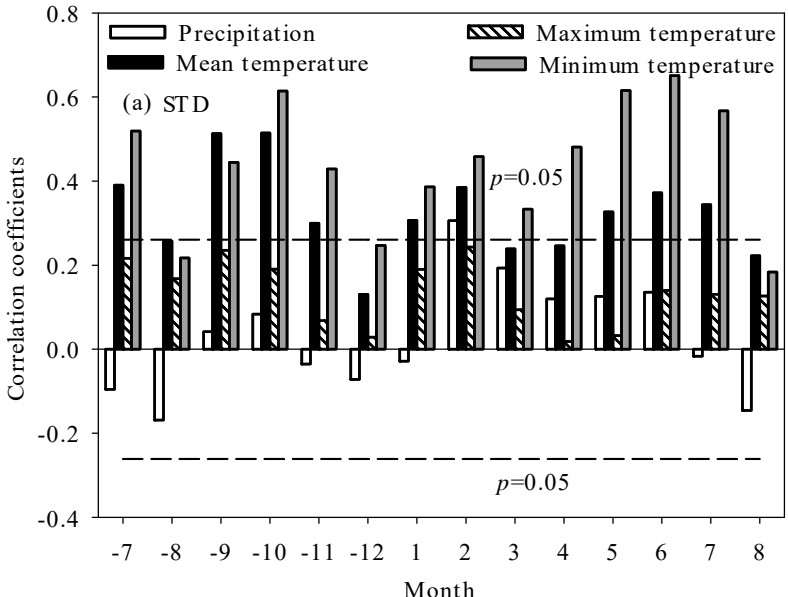

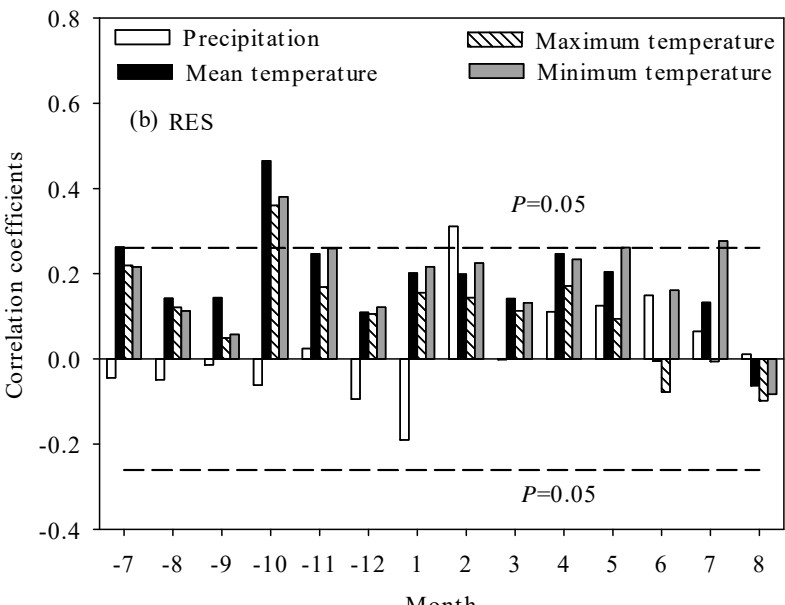

**Fig. 5**

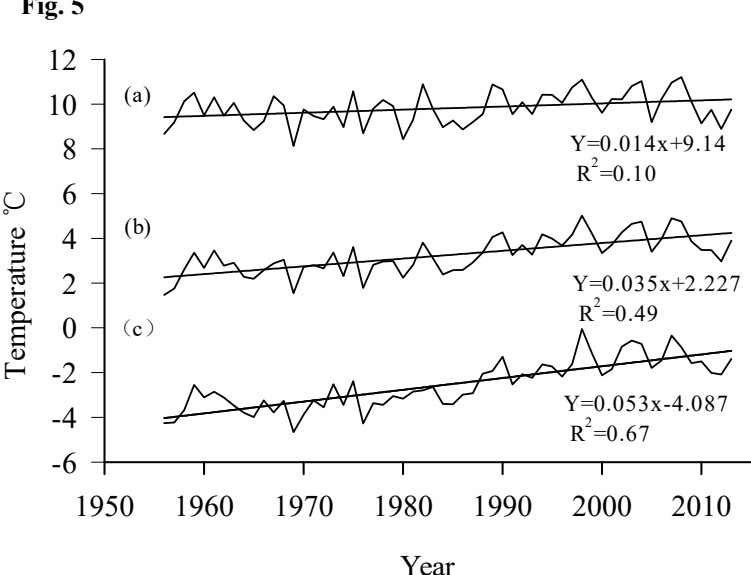

**Fig. 6**

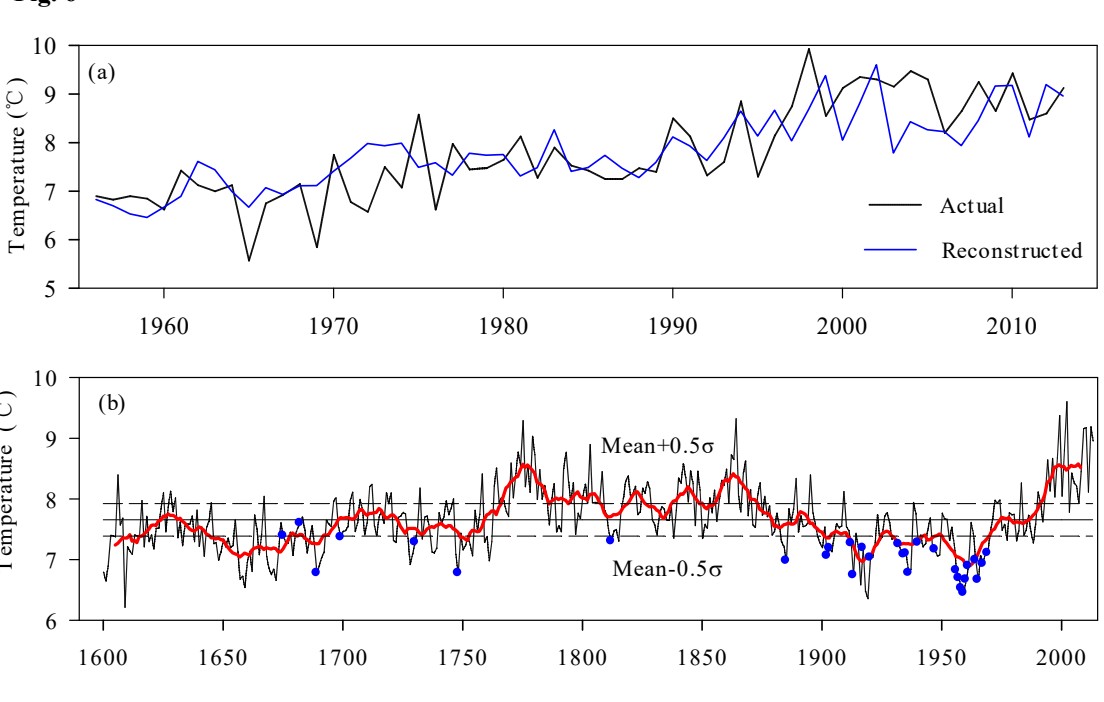

**Fig. 7**

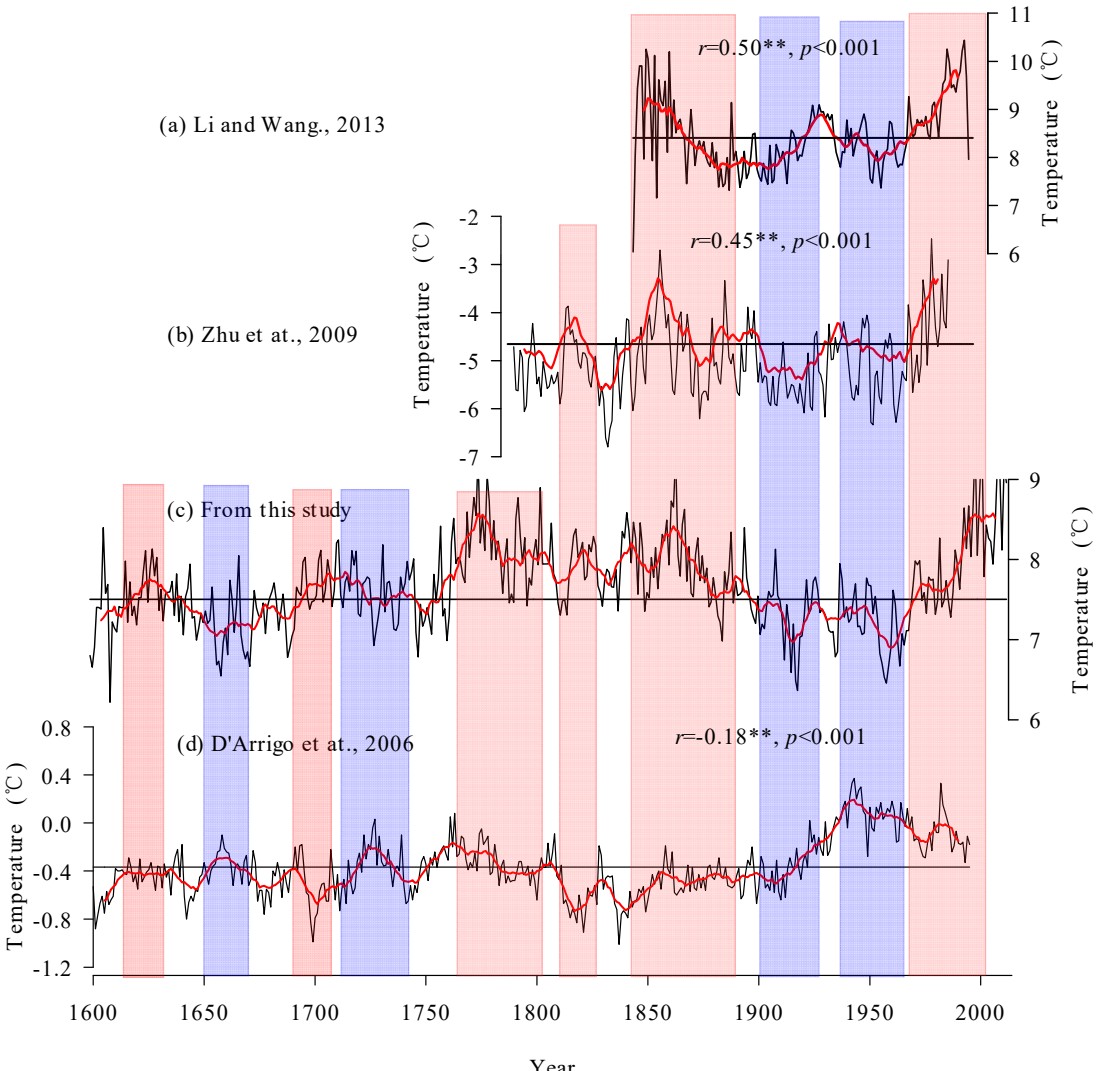

**Fig. 8**

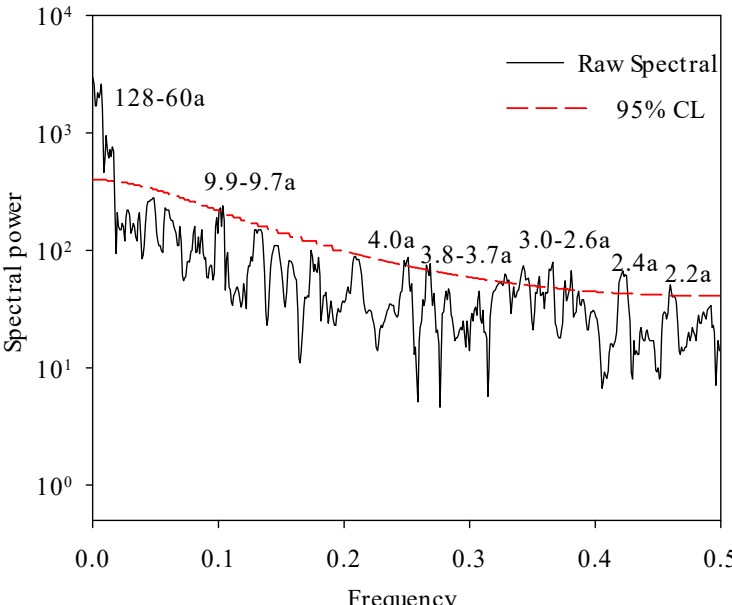