# Peer review of "A 414-year tree-ring based April-July minimum temperature reconstruction and its implications on the extreme climate events, northeast China"

_Climate of the Past, 2016_

## Referee Comment (RC1) · Anonymous Referee #1 · 29 Apr 2016

This manuscript presented a temperature reconstruction for the past 413 years in northeast China, where a long-term climatic record still remained a lack. It provides valuable information to understand past changes in temperature. I recommend a major revision by considering several issues.

1. Cambial cell division may end in end August in the study area. It has no meaning to make a correlation analysis between tree growth and climatic records until December of the current year. At most, the analysis can include until September.

2. Taking into possible influence of climatic conditions in the last year, July-December of the previous year may be included for the analysis.

[Figure]

3. The first paragraph in Introduction is too long. It is reasonable to start another paragraph line 8.

4. Line 25-26, page 2, "it is important to understand the longitudinal impacts of the climate change on forest ecosystems and human production activities in northeastern China." It is very confusing to read this sentence. This manuscript did not talk about "longitudinal impacts".

5. A scientific question may be necessary to be presented in the end of Introduction.

6. Line 30: "our new temperature record not only furthers the tree-ring series in north-eastern China". It has problem in grammar.

7. A map is necessary to show your study areas.

8. In order to show low-frequency signals, the author is better to test RCS detrending.

9. For "Climate-radial growth relationship" in page 4, it is necessary to re-organize the sentences. It will better to explain why the minimum temperature rather than the maximum temperature is crucial to determine the growth, why April-July is important? In your research site, how about the minimum temperature in April-July? It may be estimate by lapse rate along the elevation. It seems to be not very meaning to explain that high mean April-July minimum temperature reduce tree growth by inhibiting tree respiration.

10. Tree may not start growth in April in your study areas.

11. Line 5-7, page 5, it is no meaning to explain a lag effect of climate conditions in Autumn.

12. Fig 2, there is a low EPS period from 1660 to 1730. The sample depth is not enough before 1730.

13. The cold period from 1914-1922 is different with other reconstructions.

---

## Referee Comment (RC2) · Anonymous Referee #2 · 26 May 2016

Long tree-ring chronology is always exciting. I'm happy to see this manuscript developed a 413-year-long tree-ring chronology in northeast China, the longest one so far in this region. Following the basic procedure of dendroclimatology, this manuscript reconstructed the April-July minimum temperature. However, this manuscript failed to detect the driving mechanism of April-July minimum temperature variation. The periodicity analysis revealed cycles similar to sunspot activity cycles. It only means solar forcing likely play a crucial role in past climate change in the Laobai Mountain region. To support such ideal, more evidences are needed (such as comparison with the sunspot series or so). The periodicity analysis along is far from enough. Under the comprehensive consideration, I don't think this work is good enough to meet the high quality of the

journal (Climate of the Past) at current situation. I provide my personal concerns in the following part, hoping it will be helpful to the further progress of it.

Major concerns 1. It's impressive that the authors collected 54 cores from 31 trees in the studied area, and all the cores are used and successfully cross dated. The standard tree-ring chronology extended from 1600 to 2013, and lucky enough, EPS>0.85 also starts from 1600 (5 cores). However, the fact is that the core number during 1600-1650 is less than 5 (Fig. 2a). Please check this inconformity. Moreover, the quality of the chronology during around 1670-1710 is low because both EPS and Rbar decrease sharply. For the above reasons, I have to doubt the starting year of the reliable chronology. 2. Why do you deal Xt with ln ($Y=2.728\ln(X_t)+7.812$)? What's the philosophy behind it? I never see such kind of transfer function in dendroclimatology. 3. In Fig. 4a, the year to year (high-frequency) variations of the reconstruction and actual April-July MMT didn't match well. The high correlation (0.757) may be caused by similar trends. This is the biggest problem of this manuscript. What's the direct correlation coefficient between tree rings and April-July minimum temperature? Did you calculate the 1st-difference correlation coefficient between them? Therefore, the following discussions (especially the extreme cold years in Fig. 4b) are meaningless and unconvincing. 4. Table 1 indicates that "the autocorrelation order 1" is 0.75, thus except for the current year climatic records, the previous year climatic records should also be included in the climate-radial growth relationship. 5. When you do the climate-radial growth relationship analysis, current November and December shouldn't be considered. Because the annual frost-free period in the studied area is approximately 90-110 days (page 3, line 17), which means the growth season is very short. So the tree-ring width almost stops expansion in November and December. If you consider these months, please give convincing reasons. The explanation in line 5-7 in page 5 is not suitable. 6. Theoretically, it's unreasonable to compare this temperature reconstruction (April-July) with the October temperature by Yin et al. (2009), and the February-April temperature in Changbai Mountains (Zhu et al., 2009) (Fig. 5), which was influenced by the East Asian Winter Monsoon. 7. What's your definition of Little Ice Age (LIA)? According to the general

definition of LIA, the period before 1850 of this reconstruction belongs to LIA. Except for the temperature during 1605-1681 was very low, the other periods before 1850 was not so cold. Furthermore, the comparison with Northern Hemisphere temperature (NHT) (Fig. 5) is not so good. NHT (Wilson et al., 2007) showed evident increasing trend since around 1810, while this temperature reconstruction doesn't show such direct warming trend. The temperatures during most time of 19th even had opposite phase to NHT. 8. CE is a more rigorous parameter than RE in split-period calibration and verification analyses, please offer this parameter in table 2. Minor concerns 1. A map showing the general location of sample site and meteorological station is useful in helping the readers get an intuitive understanding of this work. 2. The general information of the sampled species in this manuscript should be given. It will be helpful for the understanding the following climate-growth relationships. 3. Detailed information of sampling site (e.g. longitude, latitude, main vegetation types) is needed. 4. I don't agree that 1684-1690 is a cold period and 1787-1793, 1795-1801 and 1803-1808 are warm periods (Table 3, Fig. 4b). 5. The time span in Table 1 is 1600-2014. Should it be 1600-2013? 6. 1600-2013 is 414 year, not 413 year. 7. The percentage of references during recent 5 years, especially during recent 3 years is too low.

---

## Referee Comment (RC3) · Anonymous Referee #3 · 23 Jun 2016

The manuscript from Lyu et al., presents a 413-year-long Korean pine (Pinus koraiensis) tree-ring chronology for the Laobai Mountain (northeast China). Despite the fact that the paper provides valuable information to understand past changes in temperature in this region, I have several major concerns that prevent me to give a positive assessment of the draft for publication in climate of the past. Hereafter the major points:

Major concerns. 1. It is strange for me to see that radial tree-growth was only correlated to minimum temperature for the period from April to July. I would expect to test a wider period from e.g April to August or April to September for the reconstruction. 2. I do not understand the use of a non linear model to reconstruct the April-July MMT. What is the added-value of this model? Most of the dendroclimatological studies are

based on linear models. To my knowledge logarithmic transfer function are not used for climate reconstruction. The authors should explain in detail the reasons why they chose this type of transfer function and the "biological reality" supported by this model. 3. The comparison between the newly developed reconstruction and other regional and hemispheric reconstructions is not sufficient and only concerns the last decades (p6, L7-23). This weakness does not allow to have a clear idea about the reliability of the new reconstruction. 4. The sections 3.4 and 3.5 respectively dedicated to frost disaster events and to the analysis of periodicities in the newly developed reconstruction are too weak and failed to prove (i) that trees properly recorded extreme events and (ii) to properly demonstrate the impacts of solar cycles on temperatures fluctuations in north-eastern China. 5. I would recommend a strong revision of the results and discussion section as several statements remain insufficiently demonstrated (see comments below). In addition, I would also recommend a careful revision of the language by a native speaker.

Minor concerns. 1. P2. L19-22. It would be very valuable for the paper, and especially for readers that are not familiar with recent tree-ring developments, to add a map with the location of the study sites as well as other available chronologies for China. 2. P2. L18-19. "However, tree-ring series were rarely used to reconstruct past climate (especially temperature) in this area because of the exceptional hydrothermal conditions." Could you assess the impact of hydrothermal conditions on radial growth in this region? 3. P2. L20-25. In my opinion, the necessity for a new reconstruction in this area is insufficiently explained. Please consider rephrasing this section. 4. P2. "Therefore, our new temperature record not only furthers the tree-ring series in northeastern China and provides new evidence for regional impacts of past climate variability and changes". This sentence is not clear, please consider rephrasing. 5. P2. L25-26 "it is important to understand the longitudinal impacts of the climate change on forest ecosystems and human production activities in northeastern China." This sentence is very confusing. 6. P3. L3. I strongly recommend to add a location map for the study sites. 7. P3. L20. "Tree-ring samples were obtained from the south slope of

Laobai Mountains along an elevational gradient from 950 to 1050 m". It is strange to find temperature-sensitive trees at low altitude, far from the timberline, especially on a south-facing slope. Please provide more detail here to explain how minimal temperatures could be a limiting factor at such altitude for tree-growth. 8. P3. L31-32. "A standard chronology, which preserves more low frequency signals than other chronologies, was used in the subsequent analyses". Accounting for the detrending method used in this study, the negative exponential curve, it is difficult to understand why this detrended chronology would preserve more low frequency. 9. P4. L2-3 I am really surprised that EPS could exceed 0.85 with a sample depth of only 5 trees. 10. P4. L17-18 "and the reduction of error test (RE), and product means test (PMT) are the tools used to verify the results". Please provide the coefficient of efficiency (CE) which is usually used to indicate the significance of the model skill on the verification period. 11. P4. L24-25. "Climate-growth response function analysis showed that the standard chronology was positively correlated with the mean minimum temperatures from January to December in current year". Why did you test such a long period as the growing season is probably limited to April to September? 12. P4. L25-26. "This means that cool or warm conditions are favored for the Korean Pine growth in this area." This sentence is not clear please consider rephrasing. 13. P5. L2-3. "Second, a crucial growth period of the Korean pine is from April to July." Why only April-July and not April-August or April-September, the periods usually used for reconstructions in these regions? did you test all the possible combinations of regressors? In this case, could you please provide an additional table with these analyses? These analyses are even more crucial as the authors state that " the photosynthesis still occurred during autumn, when it is generally the end of growing season; the lower mean minimum temperature reduced the tree respiration, allowing for more photosynthetic products to be stored, thus creating favorable conditions for subsequent tree growth". 14. P5. L25-28. "The longest cold period lasted from 1605 to 1681 AD (77 years), with an average temperature of 1.04 °C below the mean value". During most of this period, EPS<0.85, it is thus difficult to consider the reconstruction as reliable during the first part of the 17th century. 15.

P5. L31. "were also consistent with other results of the tree-ring reconstructions in northeastern China (Shao and Wu, 1997; Yin et al., 2009; Wang et al., 2012)". Please provide a figure in order to highlight this consistency between reconstructions. 16. P5. L33. "In addition, the two cold periods of 1605-1681 and 1684-1690 were fully consistent with the Maunder minimum (1620-1710), an interval of decreased solar ir- radiance (Bard et al., 2000)". The Mauder minimum is usually defined as the coldest period of the LIA that extends from ∼1645 to 1715. 17. P6. L4-6 "The Little Ice Age in the 17th century and the rapid warming during the mid-19th and late 20th century in northeastern China had been well recorded in this series, suggesting that this se- ries had a good regional representativeness of temperature variations in northeastern China". This is insufficiently demonstrated, please provide more details here. 18. P6. L12-16. "All of the temperature series exhibited significantly low temperatures during the 1950s-1970s, which coincided with a slight decrease in sun activity from AD 1940- 1970 (Beer et al., 2000) (Fig. 5). Another notable feature was that all of the curves showed a sharp increase from 1980, and the peak values appeared in the late 1990s and early 2000s, 15 which was consistent with the reports from the Intergovernmental Panel on Climate Change (IPCC, 2007)" Why did the authors only focus on compar- isons for the last decades ? I would expect comparisons between reconstructions for the last 400 hundred years. 19. P6. L18-20. "Additionally, three northeastern tem- perature reconstruction series showed that some cold/warm years were not analogous due to the differences in the reconstruction parameters (e.g., temperature subdivisions into average temperature, minimum temperature, maximum temperature, etc.) and habitat conditions in different sampling areas". Neither these chronologies nor their relations with climatic variables are presented here. In this context, we should only trust the authors. . . 20. P6. L27-28 "The evidence from historical documents shows that cold damage or frost disaster events have been occurring in Heilongjiang Province since 1675 (Sun et al., 2007)." Here again, I would expect a year-by-year comparison between historical archives and the tree-ring reconstruction, at least for the extreme years detected in both dataset in order to demonstrate unambiguously that tree-rings

from Korean pines properly recorded past climatic extremes.

---

## Author Comment (AC1) · 20 Jul 2016

Response to Anonymous Referee 1: We appreciate the valuable suggestions and constructive comments on the manuscript from the anonymous reviewer. These comments are very helpful for revising and improving our MS. Based on these comments, careful revisions have been made to the MS. The revisions made to the MS detailed below.

Comments: 1. Cambial cell division may end in end August in the study area. It has no meaning to make a correlation analysis between tree growth and climatic records until December of the current year. At most, the analysis can include until September.

The authors' response: Comment accepted. Months from the previous July to current August were selected for the analysis of the relationship between the climatic factors and the Korean pine growth (Fig. 1). Furthermore, the climate data included total monthly precipitation, mean maximum temperature, mean temperature, and mean minimum temperature. Months from the previous July to current August were selected for the analysis of the relationship between the climatic factors and the Korean pine growth.

2. Taking into possible influence of climatic conditions in the last year, July-December of the previous year may be included for the analysis.

The authors' response: Comment accepted. In order to consider the last year of climate impact on tree growth, July-December of the previous year has been included for the analysis. (Fig. 1) Relationships between the STD and RES chronology of the Dunhua monthly climate data were shown in Fig. 1. Correlations between the tree-ring chronologies and monthly climate data showed temperatures was more crucial to Korean pine growth compared with precipitation. In contrast, correlations between Korean pine chronologies and mean minimum temperature were positive and higher than those for maximum and mean temperature. Fig. 1 showed that the significant correlation months between STD chronology and mean minimum temperature disappeared or poorly correlated for the RES chronology. This suggests that the STD chronology represents the minimum temperature signals in low frequency, but not at high frequency. In addition, different month combinations were also considered. The best-correlated three temperature months were then selected for temperature reconstruction (Table 1). The highest correlation (r=0.757, p<0.0001) was found between STD chronology and April-July mean minimum temperature (MMT). Further, it is generally accepted that extreme temperature commonly, though not always, limits tree growth at treeline or at high latitudes forest, especially spring or early summer minimum temperature (Körner and Paulsen, 2004; Porter et al., 2013; Wilson and Luchman, 2002; Yin et at., 2015). Moreover, Tmax, Tmean and Tmin during the observed period of 1956-2013 shown in Fig. 5 illustrated the similar inter-annual variations (not shown here), while the innone

crease trend of Tmin is much higher than Tmean and Tmax, especially after 1976. This phenomenon is consistent with Karl et al. (1993), Ren et al. (1998) and Tang et al. (2005), which suggested that the global warming over past decades is mostly owing to the faster rise of night or minimum temperatures. Northeastern China is warming in similar ways. Based on the correlation between the STD chronology and the climate data, we found that compared to the maximum temperature and mean temperature, the minimum temperature (especially for April-July) plays a more important role in limiting the annual radial growth of Korean Pine in Laobai Mountain. This also means that warm and wet conditions are suitable for Korean Pine growth in this area. This may result from two reasons: First, the sampled site was located at higher elevation close to the upper limit of Korean pine distribution, which may have caused more sensitive tree growth in relation to temperature (Szeicz and MacDonald, 1995; D'Arrigo et al., 2009; Li et al., 2011; Yu et al., 2011; Flower and Smith, 2012). Early in the growing season, higher mean minimum temperatures can prevent frost damage, which is more conducive to form a wider ring (Wu, 1990; Akkemik, 2000; Makinen et al., 2003). In addition, higher nighttime temperature could promote the tree respiration and enhance the physiological activity, thereby producing more auxin, promoting cell enlargement, and forming a wider ring in the growing season (Fritts et al., 1976). As the climate warms in northeastern China, trees could carry out photosynthesis at the early stage of the growing season, higher minimum temperature is conducive to produce more auxin, promote photosynthesis rate and increase the nutrient accumulation. Therefore, we find that Korean pine tree-ring width is positively correlated with temperature. Second, we also find that a crucial growth period of the Korean pine is from April to July. During this period, the temperature could have direct effects on photosynthesis rate, cambium activity, and respiration efficiency, etc., all of which affect tree-ring width (Li et al., 2000; Yu et al., 2011).

3. The first paragraph in Introduction is too long. It is reasonable to start another paragraph line 8.

The authors' response: Comment accepted. The structure of the first paragraph is re-adjusted in Introduction. Global climate change presents major challenges for humans and the natural systems that provide ecosystem services. Consequently, it is urgent to better understand the climate change and its forcing mechanisms. Instrumental records are typically less than 100 years and often less than 50 years in most areas of the world. It is necessary to put the present climate regime in context with the long-term perspectives, which forces a reliance on natural proxy records to reconstruct the past climate. Tree rings have been widely applied in global climate change studies and paleoclimate reconstructions at both regional and global scales because they offer accurate and continuity temporal record as well as they are widespread and easily replicated (Corona et al., 2010; Bouriaud et al., 2014; Kress et al., 2014).

4. Line 25-26, page 2, "it is important to understand the longitudinal impacts of the climate change on forest ecosystems and human production activities in northeastern China." It is very confusing to read this sentence. This manuscript did not talk about "longitudinal impacts".

The authors' response: Comment accepted. The sentence was modified as follow: it is important to understand the impacts of climate change on forest ecosystems and human production activities in northeastern China.

5. A scientific question may be necessary to be presented in the end of Introduction.

The authors' response: Comment accepted. In the end of Introduction has been presented a scientific question. Therefore, there is a demand for higher-quality climate reconstructions in a greater number of areas over longer periods and a larger group of climatic indicators for verification in this region. For this reason, more information of regional past climate variations registered in a long-term tree-ring series is needed, and it is important to understand the impacts of climate change on forest ecosystems and human production activities in northeastern China. Currently, a significant climate warming (mainly is the minimum temperatures) occurred in northeastern China since

the 1980s. However, there still remains a lack of long-term climatic record (at least more than 300 years) in northeast China to explore what is the temperature regime in the past one or half thousand years. A new temperature reconstruction for the region can help answer the question, "is the current warming in northeast China unprecedented?"

6. Line 30: "our new temperature record not only furthers the tree-ring series in northeastern China". It has problem in grammar.

The authors' response: Comment accepted. The sentence was modified as follow: Our new minimum temperature record in northeastern China provides a new evidence of past climate variability, and can be used to predict the climate trend in the future in northeast China.

7. A map is necessary to show your study areas.

The authors' response: Comment accepted. A map and a panoramic photo was added to clearly show our study area. Details see Fig. 2.

8. In order to show low-frequency signals, the author is better to test RCS detrending.

The authors' response: Thank you for this suggestion. In order to obtain low-frequency signals, we used the regional curve standardization (RCS) method to process the tree-ring series during the chronology development. The following figure showed that the RCS chronology and STD chronology display similar patterns of variation at low-frequency (Fig. 3). In addition, some cores did not pass the tree center, so it was not easy estimate the rings that was missed. Further, there are many issues with using RCS on living trees. For these reasons we used the STD chronology for analysis and we took care in dealing with the variation of the reconstructed series at low frequencies.

9. For "Climate-radial growth relationship" in page 4, it is necessary to re-organize the sentences. It will better to explain why the minimum temperature rather than the maximum temperature is crucial to determine the growth, why April-July is important?

[Figure]

In your research site, how about the minimum temperature in April-July? It may be estimate by lapse rate along the elevation. It seems to be not very meaning to explain that high mean April-July minimum temperature reduce tree growth by inhibiting tree respiration.

The authors' response: Comment accepted. We have rephrased this section of climate-radial growth relationship (Please see lines 8-30, page 5 and lines 1-12, page 6 in the text). It was explained that the minimum temperature rather than the maximum temperature is crucial to determine the growth. In our study site, we estimate the minimum temperature in April-July (4.96 °C) by lapse rate (0.6 °C Âů100m-1) along the elevation, which is higher than the sap flow of Korean pine for temperature (4.5 °C). Therefore, the April-July minimum temperature plays a crucial important role in limiting the annual radial growth of Korean Pine in Laobai Mountain.

10. Tree may not start growth in April in your study areas.

The authors' response: Thank you for this suggestion. It is generally accepted that the growth of Korean pine starts from the mid- to late-April, when the activity of vascular cambium starts. April is the early stage of the growing season for Korean pine, and the main growing period is from May to August in our study area. As climate warms in northeastern China, trees seem to carry out photosynthesis at the early stage of growing season. Higher minimum temperature is conducive to produce more auxin, promote photosynthesis rate and increase the nutrient accumulation, thus, tree-ring width is positively correlated with temperature. In addition, when the mean minimum temperature in April is higher, the trees begin to grow ahead of time and come into the growing season in advance, which means that it is possible that an extended growing season is advantageous for the radial growth of trees (Wu, 1990; Akkemik, 2000; Makinen et al., 2003). Therefore, April is an important month of Korean pine growth in this area, and should be taken into account in this reconstruction.

11. Line 5-7, page 5, it is no meaning to explain a lag effect of climate conditions in

Autumn.

The authors' response: Comment accepted. The lag effect of climate conditions in Autumn was delete.

12. Fig 2, there is a low EPS period from 1660 to 1730. The sample depth is not enough before 1730.

The authors' response: Comment accepted. We went to the sampled site again on May 16, 2016 again. A total of 17 cores from 10 living trees were sampled again in the same study area. Then, a total 71 cores from 41 trees was used to develop the chronology. Therefore, the sample depth is better than before since 1630 (EPS>0.8). A generally acceptable threshold of the EPS was consistently greater than 0.85 from AD 1660 to 2015 (eleven trees) (Fig. 4).

13. The cold period from 1914-1922 is different with other reconstructions.

The authors' response: We explored nearby regions and disagree with this statement. The cold period from 1914-1922 is consistent with the results of nearby tree-ring reconstruction in the Changbai Mountains and Xiaoxing'an Mountains (Wang et al., 2012; Zhu et al., 2015). In addition, this cold period is also consistent with the severe freezing period of 1909-1918 in Heilongjiang Province (Sun et al., 2007), and 1920 is an extremely cold damage event (Sun et al., 2007).

Once again, thank you very much for your comments and suggestions. Best Regards, Shanna Lyu, on behalf of all co-authors

Please also note the supplement to this comment:
http://www.clim-past-discuss.net/cp-2016-38/cp-2016-38-AC1-supplement.pdf

[Figure]

Fig. 1 Correlations between the monthly mean meteorological data (including mean temperature, mean maximum temperature, mean minimum temperature, and total precipitation) from Dunhua meteorological station (1956-2013) and (a) the STD chronology and (b) RES chronology, respectively. The dashed horizontal line represents the 95 % confidence limit.

[Figure]

Fig. 2 Map of the sampling site, compared temperature series, nearly temperature and meteorological station in northeastern China. The photo showed the sampled site in Laobai Mountain and the remarkable vertical vegetation distribution along altitude changes.

[Figure]

Fig. 3 Comparison of the STD and RCS chronologies in Laobai Mountains

Fig. 4 Variations of (A) the STD chronology and sample depth, and (B) the expressed population signal (EPS) and average correlation between all series (Rbar) from 1600 to 2015.

Table 1. Correlation coefficients between the STD chronology and the climate data of different month combinations during the common period of 1956–2013.

| Months | $T_{mean}$ | $T_{min}$ | $T_{max}$ |
|--------|--------|--------|--------|
| c4-c7 | 0.577** | 0.757** | 0.177 |
| c4-c8 | 0.557** | 0.717** | 0.183 |
| c4-c9 | 0.599** | 0.726** | 0.217 |
| c5-c7 | 0.556** | 0.749** | 0.198 |
| c5-c8 | 0.522** | 0.691** | 0.198 |
| c5-c9 | 0.587** | 0.709** | 0.236 |
| c6-c8 | 0.447** | 0.634** | 0.199 |
| c6-c9 | 0.535** | 0.671** | 0.241 |
| p7-c8 | 0.586** | 0.682** | 0.230 |

* Significant at the 0.05 level (two-tailed). ** Significant at the 0.01 level (two-tailed).

---

## Author Comment (AC2) · 21 Jul 2016

Response to Anonymous Referee 2:

Thank you very much for your constructive comments on our manuscript, including major concerns and minor concerns. These comments are very valuable and helpful for revising and improving our MS, which also play an important guiding role in our research. We accepted most of your comments and made correction carefully.

Major concerns: 1. It's impressive that the authors collected 54 cores from 31 trees in the studied area, and all the cores are used and successfully cross dated. The standard tree-ring chronology extended from 1600 to 2013, and lucky enough, EPS>0.85

also starts from 1600 (5 cores). However, the fact is that the core number during 1600-1650 is less than 5 (Fig. 2a). Please check this inconformity. Moreover, the quality of the chronology during around 1670-1710 is low because both EPS and Rbar decrease sharply. For the above reasons, I have to doubt the starting year of the reliable chronology.

The authors' response: Comment accepted. We went to the sampled site again on May 16, 2016. A total of 17 cores from 10 living trees were sampled again near the same study area. Then, a total 71 cores from 41 trees was used to develop the chronology. Therefore, the sample depth is better than before since 1630 (EPS>0.8). A generally acceptable threshold of the EPS was consistently greater than 0.85 from AD 1660 to 2015 (eleven trees) (Fig. 4).

2. Why do you deal Xt with ln (Y=2.728 ln (Xt)+7.812)? What's the philosophy behind it? I never see such kind of transfer function in dendroclimatology.

The authors' response: At first, both the logarithmic and linear functions are good to build the reconstruction equation. Because the variance explained by the logarithmic function is better than linear function, we choose the logarithmic function in the previous MS. Now, we accepted your suggestion: the transfer function is modified as follows: Y = 2.987Xt+ 4.829.

3. In Fig. 4a, the year to year (high-frequency) variations of the reconstruction and actual April-July MMT didn't match well. The high correlation (0.757) may be caused by similar trends. This is the biggest problem of this manuscript. What's the direct correlation coeffiňA̧cient between tree rings and April-July minimum temperature? Did you calculate the 1st-difference correlation coeffiňA̧cient between them? Therefore, the following discussions (especially the extreme cold years in Fig. 4b) are meaningless and unconvincing.

The authors' response: Thank you for this suggestion. As shown in Fig. 3, the amplitude of the STD chronology was larger than RES chronology in low-frequency variability, indicating that STD chronology preserved more low-frequency signals and RES chronology reflected high-frequency signals. The mean sensitivity of the RES chronology was larger than STD chronology, which quantitatively illustrated that the RES chronology exhibits more high-frequency climate information than the STD chronology. In addition, Fig. 1 showed that the significant correlated months between the STD chronology and mean minimum temperature disappeared or poorly correlated for the RES chronology, suggesting that the STD chronology contained minimum temperatures only share low-frequency temperature variability, but not high-frequency temperature variability. Further, the first difference correlation (not shown) between the STD chronology and temperature did not exceed the 95% confidence level, which confirms that this regressed equation may be better to capture the low-frequency variability rather than high-frequency variability. Besides, the correlation coefiňチcient between the first-order difference series of the actual and reconstructed values is not significant at the 0.05 level (r=0.12, p>0.05). Therefore, this reconstructed minimum temperature series is more consistent with the observed series at low-frequency variability, which only represents the warm/cold variability at low frequencies in this region.

4. Table 1 indicates that "the autocorrelation order 1" is 0.75, thus except for the current year climatic records, the previous year climatic records should also be included in the climate-radial growth relationship.

The authors' response: Comment accepted. The previous year climatic records have been included in the climate-radial growth relationship. Months from the previous July to current August were selected for the analysis of the relationship between climatic factors and Korean pine growth (Fig. 1).

5. When you do the climate-radial growth relationship analysis, current November and December shouldn't be considered. Because the annual frost-free period in the studied area is approximately 90-110 days (page 3, line 17), which means the growth season is very short. So the tree-ring width almost stops expansion in November and December. If you consider these months, please give convincing reasons. The explanation in line

5-7 in page 5 is not suitable.

The authors' response: Comment accepted. We have rephrased this section of climate-radial growth relationship. Climate-growth response function analysis showed that the STD chronology was positively correlated with the mean minimum temperatures from the previous July to current August (Fig. 1). Relationships between the STD and RES chronologies and monthly climate data in Dunhua were shown in Fig. 1. Results showed that temperatures were more crucial to Korean pine growth compared with precipitation. In contrast, the correlation coefficients between Korean pine chronologies and mean minimum temperature were positive and higher than those for maximum and mean temperature. The significant correlation months between STD chronology and mean minimum temperature disappeared or poorly correlated for the RES chronology. This indicated that the STD chronology just recorded the minimum temperature signals in low frequency, but not high frequency temperature variability. In addition, different month combinations were also considered. The best-correlated three temperature months were then selected for temperature reconstruction (Table 1). The highest correlation coefficient (r=0.757, p<0.0001) was found between STD chronology and April-July mean minimum temperature (MMT). It is generally accepted that extreme temperature limits tree growth at treeline or at high latitudes forest, especially spring or early summer minimum temperature (Körner and Paulsen, 2004; Porter et al., 2013; Wilson and Luchman, 2002; Yin et at., 2015). Moreover, Tmax, Tmean and Tmin during the observed period of 1956-2013 shown in Fig. 5 illustrated the similar inter-annual variations, while the increase trend of Tmin is much higher than Tmean and Tmax, especially after 1976. This phenomenon is consistent with Karl et al. (1993), Ren et al. (1998) and Tang et al. (2005), which suggested that the global warming over past decades is mostly owing to the faster rise of night or minimum temperatures and the warming in northeastern China is like that. Based on the correlation between the STD chronology and the climate data, we found that compared to the maximum temperature and mean temperature, the minimum temperature (especially for April-July) plays a more important role in limiting the annual radial growth

of Korean Pine in Laobai Mountain. This also means that warm and wet conditions are suitable for Korean Pine growth in this area. This may result from two reasons: First, the sampled site was located at higher elevation close to the upper limit of Korean pine distribution, which may have caused more sensitive tree growth in relation to temperature (Szeicz and MacDonald, 1995; D'Arrigo et al., 2009; Li et al., 2011; Yu et al., 2011; Flower and Smith, 2012). In early growing season, higher mean minimum temperature can defense frost damage, thus is more conducive to form a wider ring (Wu, 1990; Akkemik, 2000; Makinen et al., 2003). In addition, higher nighttime temperature could promote the tree respiration and enhance the physiological activity, thereby producing more auxin, promoting cell enlargement, and forming a wider ring in the growing season (Fritts et al., 1976). As the climate warming in northeastern China, trees could carry out photosynthesis at the early stage of the growing season, higher minimum temperature is conducive to produce more auxin, promote photosynthesis rate and increase the nutrient accumulation. Therefore, Korean pine tree-ring width is positively correlated with temperature. Second, a crucial growth period of the Korean pine is from April to July. During this period, the temperature could have direct effects on photosynthesis rate, cambium activity, and respiration efficiency, etc., all of which affect tree-ring width (Li et al., 2000; Yu et al., 2011).

6. Theoretically, it's unreasonable to compare this temperature reconstruction (April-July) with the October temperature by Yin et al. (2009), and the February-April temperature in Changbai Mountains (Zhu et al., 2009) (Fig. 5), which was influenced by the East Asian Winter Monsoon.

The authors' response: Comment accepted. The temperature series by Yin et al. (2009) was removed from this comparison, but the series by Zhu et al. (2009) was keep because the two sites are close and both contain April, and the most important one is that they show very similar variation patterns. The reconstructed temperature series of Changbai Mountains (Zhu et al., 2009) was significantly positively (r = 0.454**, p < 0.01) correlation. Therefore, it seems reasonable to compare this temperature reconstruction (April-July) with the February-April temperature in Changbai Mountains (Zhu et al., 2009). In addition, we provide an additional Table 1 with these analyses of various month combinations. It was showed that the photosynthesis still occurred during autumn in our study site, when it is generally the end of growing season; the lower mean minimum temperature reduced the tree respiration, allowing for more photosynthetic products to be stored, thus creating favorable conditions for subsequent tree growth (Gao et al., 2011; Wang et al., 2011). Thus, it's reasonable to compare this temperature reconstruction (April-July) with the October temperature by Yin et al. (2009).

7. What's your definition of Little Ice Age (LIA)? According to the general C2CPD Interactive Comment Printer-friendly version Discussion paper definition of LIA, the period before 1850 of this reconstruction belongs to LIA. Except for the temperature during 1605-1681 was very low, the other periods before 1850 was not so cold. Furthermore, the comparison with Northern Hemisphere temperature (NHT) (Fig. 5) is not so good. NHT (Wilson et al., 2007) showed evident increasing trend since around 1810, while this temperature reconstruction doesn't show such direct warming trend. The temperatures during most time of 19th even had opposite phase to NHT.

The authors' response: Thank you for this suggestion. A significant negative correlation (r = -0.179**, p < 0.01) between our reconstruction and the northern hemisphere temperature data (D'Arrigo et al., 2006) was also found (Fig. 6). It was widely believed that the Little Ice Age in China has three cold periods, that was the 15th century, the 17th century and the 19th century (Wang et al., 2003). The first period was relatively less obvious, and the second period was most obvious of all but different in when it begins and ends, however, the third period has some regional differences (e.g. the southern China was obvious and the northeast region and Sinkiang were the opposite) (Wang et al., 1998; Wang et al., 2003). The third Little Ice Age in 19th century was not obvious in our reconstruction, which was consistent with Wu et al. (1998) and Wang et al. (1998), and which also lead to the bad matching with Northern Hemisphere temperature. In addition, the climate was warm for the late 18th and early 19th in Heilongjiang Province (Gong et al., 1979). While the LIA is a genereal convention for a certain period, we do not think we should expect perfect phasing across regions and seasons of reconstruction. It might be that northeastern China has occasionally experienced significant departures from global trends.

8. CE is a more rigorous parameter than RE in split-period calibration and verification analyses, please offer this parameter in table.

The authors' response: Comment accepted. We have added a rigorous parameter of CE in Table 2.

Minor concerns: 1. A map showing the general location of sample site and meteorological station is useful in helping the readers get an intuitive understanding of this work.

The authors' response: Comment accepted. A map and a landscape photo was added to clearly show our study area. Details see Fig. 2.

2. The general information of the sampled species in this manuscript should be given. It will be helpful for the understanding the following climate-growth relationships.

The authors' response: Comment accepted. The general information of the sampled species has been added to the MS. Five tree species were cored in this area, but only Korean pine (Pinus koraiensis) cores were used in this study. Korean pine is a sun-loving plant (shade tolerant when it is young) and has shallow roots, widely distribute on well-drained wet mountain slopes close to the subalpine timberline where the brown forest soil is covered.

3. Detailed information of sampling site (e.g. longitude, latitude, main vegetation types) is needed.

The authors' response: Comment accepted. The detailed information of sampling site was added into the MS. The study area is located at Laobai Mountain (128°03' E,

44°06' N) , the boundary zone between Jilin and Heilongjiang provinces, and is also an ecotone between the Changbai and Xiaoxing'an Mountain. Laobai Mt. is the third highest peak in northeastern China and rises to 1650 m above sea level (a.s.l.). Almost no inhabitants live in or near the Mountain, so the forest ecosystem is preserved very well and the native vegetation remains predominantly intact (Fig. 1). Five forest vegetation types from temperate to frigid change with the altitude increase, which is Quercus mongolica broad-leaved forest below 800 m a.s.l., the mixed broadleaved Korean pine forest from 800 to 1050 m, dark conifer forest with Picea jezoensis from 1050 to 1350 m, Betula ermanii forest between 1350 and 1640 m, and Pinus pumila forest and subalpine meadow above 1640 m. Plant species is a transition from Changbai Mountain to Xiaoxing'an Mountain. Five tree species were cored in this area, but only Korean pine (Pinus koraiensis) cores were used in this study. Korean pine is a sun-loving plant and has shallow roots, widely distribute on well-drained wet mountain slopes close to the subalpine timberline where the brown forest soil is covered. The vegetation of this area is mixed broadleaved Korean pine forest dominated by Pinus koraiensis, Picea jezoensis and Abies nephrolepis as well as broadleaf tree species, such as Juglans mandshurica, Fraxinus mandshurica and Acer mono (Bu et al., 2003).

4. I don't agree that 1684-1690 is a cold period and 1787-1793, 1795-1801 and 1803-1808 are warm periods (Table 3, Fig. 4b).

The authors' response: Thank you for this suggestion. We checked it again. The cold period of 1684-1690 was consistent with cold period of 1689-1690 in Heilongjiang Province (Gong et al., 1979). The warm periods of 1787-1793, 1795-1801 and 1803-1808 were found in nearby tree-ring reconstruction series in Changbai Mountains (Wang et al., 2012; Shao et al., 1997).

5. The time span in Table 1 is 1600-2014. Should it be 1600-2013?

The authors' response: Thank you for this suggestion. The time span of STD chronology in Laobai Mountain is 1600-2015 in Table 1, but the time span of the reconstructed

minimum temperature series is 1600-2013.

6. 1600-2013 is 414 year, not 413 year

The authors' response: Comment accepted. All such errors in the MS were corrected.

7. The percentage of references during recent 5 years, especially during recent 3 years is too low.

The authors' response: Comment accepted. New reference during recent 5 years (such as Zhu et al., 2015; Wu et al., 2013; Yin et al., 2015; Porevor et al., 2013, etc.) were added into the appropriate locations of the main text.

Once again, thank you very much for your comments and suggestions.

Best Regards,

Shanna Lyu, on behalf of all co-authors

Please also note the supplement to this comment:
http://www.clim-past-discuss.net/cp-2016-38/cp-2016-38-AC2-supplement.pdf
* * *
[Figure]

Fig. 1 Correlations between the monthly mean meteorological data (including mean temperature, mean maximum temperature, mean minimum temperature, and total precipitation) from Dunhua meteorological station (1956-2013) and (a) the STD chronology and (b) RES chronology, respectively. The dashed horizontal line represents the 95 % confidence limit.

[Figure]

Fig. 2 Map of the sampling site, compared temperature series, nearly temperature series and meteorological station in northeastern China. The photo showed the sampled site in Laobai Mountain and the remarkable vertical vegetation distribution along altitude changes.

[Figure]

Fig. 4 Variations of (A) the STD chronology and sample depth, and (B) the expressed population signal (EPS) and average correlation between all series (Rbar) from 1600 to 2015.

[Figure]

**Fig. 5** (a) April-September mean minimum temperature in Dunhu reconstructed by Li and Wang

Table 1. Correlation coefficients between the STD chronology and the climate data of different month combinations during the common period of 1956–2013.

| Months | $T_{mean}$ | $T_{min}$ | $T_{max}$ |
|--------|--------|--------|--------|
| c4-c7 | 0.577** | 0.757** | 0.177 |
| c4-c8 | 0.557** | 0.717** | 0.183 |
| c4-c9 | 0.599** | 0.726** | 0.217 |
| c5-c7 | 0.556** | 0.749** | 0.198 |
| c5-c8 | 0.522** | 0.691** | 0.198 |
| c5-c9 | 0.587** | 0.709** | 0.236 |
| c6-c8 | 0.447** | 0.634** | 0.199 |
| c6-c9 | 0.535** | 0.671** | 0.241 |
| p7-c8 | 0.586** | 0.682** | 0.230 |

* Significant at the 0.05 level (two-tailed). ** Significant at the 0.01 level (two-tailed).

Table 2. Calibration and verification statistics of the reconstruction equation for the common period of 1956-2013

| Calibration | R | R$^2$ | Verification | R | Reduction of Error | Coefficient of efficiency | Sign Test | Product Mean Test |
|---|---|---|---|---|---|---|---|---|
| whole Section (1956-2013) | 0.757** | 0.573** | - | - | - | - | - | - |
| Front Section (1956-1984) | 0.414* | 0.171* | Back Section (1985-2013) | 0.632** | 0.738** | 0.446** | (20, 9)* | 4.586** |
| Back Section (1985-2013) | 0.632** | 0.400** | Front Section (1956-1984) | 0.414* | 0.738** | 0.634** | (22, 7)** | 6.099** |

* Significant at the 0.05 level (two-tailed). ** Significant at the 0.01 level (two-tailed).

---

## Author Comment (AC3) · 21 Jul 2016

Response to Anonymous Referee 3:

We are grateful for the anonymous reviewers' valuable comments on the MS. Those comments are all constructive and very helpful for revising and improving our paper, as well as the important guiding significance to our researches. We have enclosed the revised MS based on the reviewers' suggestion. The following are our responses to the reviewers:

Major concerns: 1. It is strange for me to see that radial tree-growth was only correlated to minimum temperature for the period from April to July. I would expect to test a wider

period from e.g April to August or April to September for the reconstruction.

The authors' response: Thank you for this suggestion. We provide an additional Table 1 with these analyses of various month combinations. The highest correlation (r=0.757, p<0.0001) was found between STD chronology and April-July mean minimum temperature (MMT). It is generally accepted that the sap flow of Korean pine starts from the mid- to late-April, when the activity of cambium starts. April is the early stage of the growing season for Korean pine, and the main growing season is from May to August in our study area. As climate warms in northeastern China, trees could carry out photosynthesis at the early stage of the growing season, higher minimum temperature is conducive to produce more auxin, promote photosynthesis rate and increase the nutrient accumulation. July is the rapid growth period of Pinus koraiensis, which is important for the formation of early wood. We estimate the minimum temperature in April-July (4.96 °C) by lapse rate (0.6 °C 100m-1) along the elevation, which is higher than the sap flow of Korean pine for temperature (4.5 °C) in this region. Besides, if a wider period was added, the temperature variance explained by the STD chronology will decrease sharply. Therefore, we choose a period from April to July for this reconstruction.

2. I do not understand the use of a non linear model to reconstruct the April-July MMT. What is the added-value of this model? Most of the dendroclimatological studies are based on linear models. To my knowledge logarithmic transfer function are not used for climate reconstruction. The authors should explain in detail the reasons why they chose this type of transfer function and the "biological reality" supported by this model.

The authors' response: Comment accepted. At first, both the logarithmic and linear functions are good to build the reconstruction equation, while the variance explained by the logarithmic function is better than linear function. So we choose it in the previous MS. Now, we accepted your suggestion: the transfer function is modified as follows: Y = 2.987Xt+ 4.829.

3. The comparison between the newly developed reconstruction and other regional and hemispheric reconstructions is not sufficient and only concerns the last decades (p6, L7-23). This weakness does not allow to have a clear idea about the reliability of the new reconstruction.

The authors' response: Comment accepted. A new comparison of different temperature reconstructions was done. Correlation function coefficients were calculated between our reconstructed temperature series and other temperature reconstructions and northern hemisphere temperature reconstruction. We compared our reconstructed series with two nearby tree-ring based temperature reconstructions: one is from the Dunhua by Li and Wang (2013) and another one is from the Changbai Mountains by Zhu et al. (2009). Two nearby reconstructed temperature series were significantly positively ($r = 0.499^{**}$, $p < 0.01$; $r = 0.454^{**}$, $p < 0.01$) associated with our reconstructed temperature series, respectively. Meanwhile, a significant negative correlation ($r = -0.179^{**}$, $p < 0.01$) between our reconstruction and the northern hemisphere temperature data (D'Arrigo et al., 2006) was also found (Fig. 5).

4. The sections 3.4 and 3.5 respectively dedicated to frost disaster events and to the analysis of periodicities in the newly developed reconstruction are too weak and failed to prove (i) that trees properly recorded extreme events and (ii) to properly demonstrate the impacts of solar cycles on temperatures fluctuations in north-eastern China.

The authors' response: Comment accepted. We provide an additional Table 3 which is the comparison of extreme events between historical archives and this tree-ring reconstruction. There is a year-by-year comparison between historical archives and the tree-ring reconstruction to prove that trees appear to have recorded extreme events. In addition, in order to properly demonstrate the impacts of solar cycles on temperatures fluctuations in northeastern China, we offer a comparison between reconstructed April-July temperature series and April-July sunspot numbers as a reference (Fig. 6).

5. I would recommend a strong revision of the results and discussion section as several

statements remain insufiňĄciently demonstrated (see comments below). In addition, I would also recommend a careful revision of the language by a native speaker.

The authors' response: Comment accepted. The results and discussion section were rewritten by a native English speaker.

Minor concerns:

1. P2. L19-22. It would be very valuable for the paper, and especially for readers that are not familiar with recent tree-ring developments, to add a map with the location of the study sites as well as other available chronologies for China.

The authors' response: Comment accepted. A map and a panoramic photo was added to clearly show our study area (Fig. 2).

2. P2. L18-19. "However, tree-ring series were rarely used to reconstruct past climate (especially temperature) in this area because of the exceptional hydrothermal conditions." Could you assess the impact of hydrothermal conditions on radial growth in this region?

The authors' response: Thank you for this suggestion. This region belongs to temperate continental monsoon climate. Records from the nearest meteorological station in Dunhua, the mean annual temperature is 3.3 °C from 1956 to 2013, with July (mean temperature of 20.1 °C) and January (-16.8 °C) being the warmest and the coldest month, respectively. The mean monthly minimum and maximum temperatures are -2.5 and 9.8 °C, respectively. The mean annual total precipitation is 627 mm, the majority (63.1%) of which falls during June-August. The annual frost-free period is approximately 90-110 days. The temperature starts rising in April, and May–June is a comparatively warm and dry period in this region. Although July is the hottest month of the year, moisture limited is no longer a problem because of the ample summer monsoon rainfall. In the growing season, higher temperature can improve photosynthesis rate, accumulate nutrients, thus could be more advantages for tree to form a wider ring. We

mention "the exceptional hydrothermal conditions", it is not too cold or too warm, and is not too dry and wet. So it is not easy to find which factors (temperature or precipitation) is more important to limit tree growth here. Therefore, few climate reconstructions have done here. Our site is nearly located to the upper limit of Korean pine distribution in this Mountain. So it is sensitivity to temperature and appears usable for a temperature reconstruction.

3. P2. L20-25. In my opinion, the necessity for a new reconstruction in this area is insufiňĄciently explained. Please consider rephrasing this section.

The authors' response: Comment accepted. We rephrased this section. However, tree-ring series were rarely used to reconstruct past climate (especially temperature) in this area because of the exceptional hydrothermal conditions. Several temperature-sensitive tree-ring chronologies were developed in Changbai Mountain (e.g., Shao and Wu, 1997; Zhu et al., 2009; Wang et al., 2012; Li and Wang, 2013) and Xiaoxing'an Mountain (Yin et al., 2009; Zhu et al., 2015), but almost no results were obtained for the past 250 years and few reflect low-frequency climate variations. These issues limit our understanding for a longer time scale of climate variations. The existing temperature reconstructions are far from adequate and do not satisfy the demands of scientific research. Therefore, there is a need for more high-quality climate reconstructions that cover longer periods in this region. For this reason, more information of regional past climate variations registered in a long-term tree-ring series is needed, and it is important to understand the impacts of climate change on forest ecosystems and human production activities in northeastern China.

4. P2. "Therefore, our new temperature record not only furthers the tree-ring series in northeastern China and provides new evidence for regional impacts of past climate variability and changes". This sentence is not clear, please consider rephrasing.

The authors' response: Comment accepted. The sentence was modified as follow: Our new minimum temperature record supplements existing data in northeastern China

and provides a new evidence of past climate variability. There is the potential to better understand future climatic trajectories in northeast China from these data.

5. P2. L25-26 "it is important to understand the longitudinal impacts of the climate change on forest ecosystems and human production activities in northeastern China." This sentence is very confusing.

The authors' response: Comment accepted. The sentence was modified as follow: it is important to understand the impacts of climate change on forest ecosystems and the ecosystems services provided to humans in northeastern China.

6. P3. L3. I strongly recommend to add a location map for the study sites.

The authors' response: Comment accepted. A map and a panoramic photo was added to clearly show our study area (Fig. 2).

7. P3. L20. "Tree-ring samples were obtained from the south slope of Laobai Mountains along an elevational gradient from 950 to 1050 m". It is strange to find temperature-sensitive trees at low altitude, far from the timberline, especially on a south-facing slope. Please provide more detail here to explain how minimal temperatures could be a limiting factor at such altitude for tree-growth.

The authors' response: Thank you for this suggestion. In our study area, the south as the windward slope has more rainfall, so the upper limit of vegetation distribution at southern slopes is higher than that at northern slopes, and the southern slope has some vegetation types which does not exist in northern slopes. In addition, the northeast China is a part of high latitude region. The highest peak of Laobai Mountain rises to 1650 m, which is the third highest peak in northeast China. The vegetation remains predominantly original, and which is occupied by the mixed broadleaved Korean pine forest from 800 to 1050 m and dark conifer forest with Picea jezoensis from 1050 to 1350 m. It is generally accepted that extreme temperature limits tree growth at treeline or at high latitudes forest, especially spring or early summer minimum temperature

(Körner and Paulsen, 2004; Porter et al., 2013; Wilson and Luchman, 2002; Yin et at., 2015). The sampled site was located at higher elevation and close to the upper limit of Korean pine distribution, which may be more sensitive tree growth in relation to temperature (Szeicz and MacDonald, 1995; D'Arrigo et al., 2009; Li et al., 2011; Yu et al., 2011; Flower and Smith, 2012).

8. P3. L31-32. "A standard chronology, which preserves more low frequency signals than other chronologies, was used in the subsequent analyses". Accounting for the detrending method used in this study, the negative exponential curve, it is difficult to understand why this detrended chronology would preserve more low frequency.

The authors' response: Thank you for this suggestion. ARSTAN (Cook 1985) was used to detrend and standardize cross-dated tree-ring width series into a tree-ring chronology. During this detrending process, to remove biological factors (such as age-related trends) and non-climatic variations and preserve as much low-frequency signal as possible, each ring-width series were fitted with a straight line or negative exponential function. A 67% cubic smoothing spline with a 50% cutoff frequency was further used in a few cases when anomalous growth trends occurred. The detrended data from individual tree cores were then averaged using a bi-weight robust mean to form the standard (STD), residual (RES) and autoregressive (ARS) chronologies (Cook and Kairiukstis, 1990). In addition, we compared RCS (regional curve standardization) chronology and STD chronology. The following figure showed that the RCS chronology and STD chronology display similar patterns of variation at low-frequency. Therefore, STD chronology also preserved more low-frequency signals and was used in the subsequent analyses. As shown in Fig. 3, the amplitude of the STD chronology was larger than RES chronology in low-frequency variability, indicating that STD chronology also preserved more low-frequency signals.

9. P4. L2-3 I am really surprised that EPS could exceed 0.85 with a sample depth of only 5 trees.

The authors' response: Comment accepted. We went to the sampled site again on May 16, 2016 again. A total of 17 cores from 10 living trees were sampled again in the same study area. Then, a total 71 cores from 41 trees was used to develop the chronology. Therefore, the sample depth is better than before since 1630 (EPS>0.8). A generally acceptable threshold of the EPS was consistently greater than 0.85 from AD 1660 to 2015 (eleven trees) (Fig. 4).

10. P4. L17-18 "and the reduction of error test (RE), and product means test (PMT) are the tools used to verify the results". Please provide the coefficient of efficiency (CE) which is usually used to indicate the signiïficance of the model skill on the ver-ification period.

The authors' response: Comment accepted. We have added a rigorous parameter of CE in Table 2.

11. P4. L24-25. "Climate-growth response function analysis showed that the standard chronology was positively correlated with the mean minimum temperatures from January to December in current year". Why did you test such a long period as the growing season is probably limited to April to September?

The authors' response: Comment accepted. Climate-growth response function analysis showed that the STD chronology was positively correlated with the mean minimum temperatures from previous July to current August. We provide an additional Table 1 with these analyses of various month combinations. The highest correlation (r=0.757, p<0.0001) was found between STD chronology and April-July mean minimum temperature (MMT). In addition, we estimate the minimum temperature in April-July (4.96 °C) by lapse rate (0.6 °C Âů100m-1) along the elevation, which is higher than the sap flow of Korean pine for temperature (4.5 °C). Therefore, the April-July minimum temperature plays a crucial important role in limiting the annual radial growth of Korean Pine in Laobai Mountain.

12. P4. L25-26. "This means that cool or warm conditions are favored for the Korean

Pine growth in this area." This sentence is not clear please consider rephrasing.

The authors' response: Comment accepted. The sentence was modified as follow: This also means that warm and wet conditions are suitable for the growth Korean Pine in this area.

13. P5. L2-3. "Second, a crucial growth period of the Korean pine is from April to July." Why only April-July and not April-August or April-September, the periods usually used for reconstructions in these regions? did you test all the possible combinations of regressors? In this case, could you please provide an additional table with these analyses? These analyses are even more crucial as the authors state that "the photosynthesis still occurred during autumn, when it is generally the end of growing season; the lower mean minimum temperature reduced the tree respiration, allowing for more photosynthetic products to be stored, thus creating favorable conditions for subsequent tree growth".

The authors' response: Comment accepted. We provide an additional Table 1 with these analyses of various month combinations. The highest correlation (r=0.757, p<0.0001) was found between STD chronology and April-July mean minimum temperature (MMT). It is generally accepted that the sap flow of Korean pine starts from the mid- to late-April, when the activity of cambium starts. April is the early stage of the growing season for Korean pine, and the main growing season is from May to August in our study area. As the climate warming in northeastern China, trees could carry out photosynthesis at the early stage of the growing season, higher minimum temperature is conducive to produce more auxin, promote photosynthesis rate and increase the nutrient accumulation. July is the rapid growth period of Pinus koraiensis, which is important for the formation of early wood. We estimate the minimum temperature in April-July (4.96 °C) by lapse rate (0.6 °C 100m-1) along the elevation, which is higher than the sap flow of Korean pine for temperature (4.5 °C) in this region. In addition, if more temperature months were added, the temperature variance explained by the STD chronology will decrease sharply. Therefore, we choose a period from April to July for

this reconstruction.

14. P5. L25-28. "The longest cold period lasted from 1605 to 1681 AD (77 years), with an average temperature of 1.04 °C below the mean value". During most of this period, EPS<0.85, it is thus difficult to consider the reconstruction as reliable during the first part of the 17th century.

The authors' response: Comment accepted. We went to the sampled site again on May 16, 2016 again. A total of 17 cores from 10 living trees were sampled again in the same study area. Then, a total 71 cores from 41 trees was used to develop the chronology. Therefore, the sample depth is better than before since 1630 (EPS>0.8). A generally acceptable threshold of the EPS was consistently greater than 0.85 from AD 1660 to 2015 (eleven trees) (Fig. 4). Therefore, the latter portion of the 1630-1681period (EPS>0.80) is reliable.

15. P5. L31. "were also consistent with other results of the tree-ring reconstructions in northeastern China (Shao and Wu, 1997; Yin et al., 2009; Wang et al., 2012)". Please provide a figure in order to highlight this consistency between reconstructions.

The authors' response: Comment accepted. A map and a panoramic photo was added to clearly show our study area (Fig. 2).

16. P5. L33. "In addition, the two cold periods of 1605-1681 and 1684-1690 were fully consistent with the Maunder minimum (1620-1710), an interval of decreased solar irradiance (Bard et al., 2000)". The Maunder minimum is usually defined as the coldest period of the LIA that extends from 1645 to 1715.

The authors' response: Comment accepted. The cold period of the new reconstructed series has changed. The sentence was modified as follow: In addition, the two cold periods of 1645-1677 and 1684-1691 were fully consistent with the Maunder minimum (1645-1715), an interval of decreased solar irradiance (Bard et al., 2000)

17. P6. L4-6 "The Little Ice Age in the 17th century and the rapid warming during the

mid-19th and late 20th century in northeastern China had been well recorded in this series, suggesting that this series had a good regional representativeness of temperature variations in northeastern China". This is insufiňĄciently demonstrated, please provide more details here.

The authors' response: Comment accepted. We compared our data series with two nearby tree-ring based temperature reconstruction series from Dunhua by Li and Wang (2013) and Changbai Mountains by Zhu et al. (2009). The two temperature series were significantly positively (r = 0.499**, p < 0.01; r = 0.454**, p < 0.01) correlated with our reconstructed temperature series, respectively. Meanwhile, a significant negative correlation (r = -0.179**, p < 0.01) between our reconstruction and the northern hemisphere temperature data (D'Arrigo et al., 2006) was also found (Fig. 5). The three reconstructed temperature series showed analogous cold/warm variation in mid-19th and late 20th century in northeastern China

18. P6. L12-16. "All of the temperature series exhibited significantly low temperatures during the 1950s-1970s, which coincided with a slight decrease in sun activity from AD 1940-1970 (Beer et al., 2000) (Fig. 5). Another notable feature was that all of the curves showed a sharp increase from 1980, and the peak values appeared in the late 1990s and early 2000s, which was consistent with the reports from the Intergovernmental Panel on Climate Change (IPCC, 2007)" Why did the authors only focus on comparisons for the last decades ? I would expect comparisons between reconstructions for the last 400 hundred years.

The authors' response: Thank you for this suggestion. The tree-ring series were rarely used to reconstruct past climate (especially temperature) in this area because of good hydrothermal conditions. Several temperature-sensitive tree-ring chronologies were developed in the Changbai Mountains (e.g., Shao and Wu, 1997; Zhu et al., 2009; Wang et al., 2012; Li and Wang, 2013) and Xiaoxing'an Mountains (Yin et al., 2009; Zhu et al., 2015), but almost no results were obtained for the period of over 250 years and reflected the low-frequency climate variations, which limit our understanding for a

longer time scale of climate variations. Therefore, we only focus on comparison with the temperature series for the last 200 years especially in recent decades in northeastern China. Fortunately, the northern Hemisphere temperature series (D'Arrigo et al., 2006) and historical documents can make up for this shortcoming a little bit.

19. P6. L18-20. "Additionally, three northeastern temperature reconstruction series showed that some cold/warm years were not analogous due to the differences in the reconstruction parameters (e.g., temperature subdivisions into average temperature, minimum temperature, maximum temperature, etc.) and habitat conditions in different sampling areas". Neither these chronologies nor their relations with climatic variables are presented here. In this context, we should only trust the authors. . ..

The authors' response: Thank you for this suggestion. This sentence is a bit confusing. It was modified as follow: Additionally, the three northeastern temperature reconstruction series showed that some of the cold/warm periods may not be analogous with each other, which could be caused by the differences of the reconstructed temperature variable (that is, mean temperature, minimum temperature, maximum temperature) and the habitat conditions in different sampling areas. Three cold/warm periods were inconsistent with two comparison temperature series which were also found in other nearby tree-ring reconstruction series. The cold period from 1914-1922 was consistent with the results of nearby tree-ring reconstructed temperatures in Changbai Mountains and Xiaoxing'an Mountains (Wang et al., 2012; Zhu et al., 2015). The warm periods from 1795-1807 and 1819-1826 were also consistent with that in Changbai Mountains temperature reconstructions (Wang et al., 2012; Shao et al., 1997).

20. P6. L27-28 "The evidence from historical documents shows that cold damage or frost disaster events have been occurring in Heilongjiang Province since 1675 (Sun et al., 2007)." Here again, I would expect a year-by-year comparison between historical archives and the tree-ring reconstruction, at least for the extreme years detected in both datasets in order to demonstrate unambiguously that tree-rings from Korean pines properly recorded past climatic extremes.

The authors' response: Comment accepted. We provide an additional Table 3 which is the comparison of extreme events between historical archives and this tree-ring reconstruction.

Once again, thank you very much for your comments and suggestions.

Best Regards,

Shanna Lyu, on behalf of all co-authors

Please also note the supplement to this comment:
http://www.clim-past-discuss.net/cp-2016-38/cp-2016-38-AC3-supplement.pdf

————————————————

[Figure]

Fig. 2 Map of the sampling site, compared temperature series, nearly temperature series and meterological station in northeastern China. The photo showed the sampled site in Laobai Mountain and the remarkable vertical vegetation distribution along altitude changes.

[Figure]

Fig. 3 Comparison of the STD and RCS chronologies in Laobai Mountains

[Figure]

Fig. 4 Variations of (A) the STD chronology and sample depth, and (B) the expressed population signal (EPS) and average correlation between all series (Rbar) from 1600 to 2015.

[Figure]

**Fig. 5** (a) April-September mean minimum temperature in Dunhu reconstructed by Li and Wang

[Figure]

Fig 6. Comparison between the reconstructed April-July temperature series and sunspot numbers

Table 1. Correlation coefficients between the STD chronology and the climate data of different month combinations during the common period of 1956–2013.

| Months | $T_{mean}$ | $T_{min}$ | $T_{max}$ |
|---|---|---|---|
| c4-c7 | 0.577** | 0.757** | 0.177 |
| c4-c8 | 0.557** | 0.717** | 0.183 |
| c4-c9 | 0.599** | 0.726** | 0.217 |
| c5-c7 | 0.556** | 0.749** | 0.198 |
| c5-c8 | 0.522** | 0.691** | 0.198 |
| c5-c9 | 0.587** | 0.709** | 0.236 |
| c6-c8 | 0.447** | 0.634** | 0.199 |
| c6-c9 | 0.535** | 0.671** | 0.241 |
| p7-c8 | 0.586** | 0.682** | 0.230 |

* Significant at the 0.05 level (two-tailed). ** Significant at the 0.01 level (two-tailed).

Table 2. Calibration and verification statistics of the reconstruction equation for the common period of 1956-2013

| Calibration | R | R$^2$ | Verification | R | Reduction of Error | Coefficient of efficiency | Sign Test | Product Mean Test |
|---|---|---|---|---|---|---|---|---|
| whole Section (1956-2013) | 0.757** | 0.573** | - | - | - | - | - | - |
| Front Section (1956-1984) | 0.414* | 0.171* | Back Section (1985-2013) | 0.632** | 0.738** | 0.446** | (20, 9)* | 4.586** |
| Back Section (1985-2013) | 0.632** | 0.400** | Front Section (1956-1984) | 0.414* | 0.738** | 0.634** | (22, 7)** | 6.099** |

* Significant at the 0.05 level (two-tailed). ** Significant at the 0.01 level (two-tailed).

Table 3. Cold damage or frost disaster events recorded in historical archives
of Heilongjiang Province since 1675.

| 17th century | 18th century | 19th century | 20th century |
|---|---|---|---|
| 1675 | 1730 | 1800-1801 | 1901-1903 |
| 1682 | 1746 | 1812-1813 | 1909-1915 |
| 1689 | 1749 | 1830-1832 | 1917 |
| 1699 | 1748 | 1878-1879 | 1920 |
| | 1755 | 1885 | 1925-1926 |
| | 1757 | | 1931-1932 |
| | | | 1934-1936 |
| | | | 1939-1943 |
| | | | 1947 |
| | | | 1950-1969 |
| | | | 1998-1999 |
| | | | 1971-1981 |

---

## Editor Comment (EC1) · J. Guiot (Editor) · 23 Jul 2016

Dear authors,

could submit a revised version of your paper that will be considered in the same time than your replies by the reviewers. This will shortened the review process. If you are not able to download it, please send it to the editorial@copernicus.org

Best regards

Joel Guiot, editor
* * *

---

## Author Response (AR2)

**Dear Joel and two referees,**

We carefully revised our manuscript according to your comments. We appreciate your helpful comments on our manuscript. These comments make our manuscript more perfect and accurate. All detailed revision and response are as below. Thank you so much for all your help in processing our manuscript.

Sincerely yours,

Shanna Lyu and Xiaochun Wang,
on behalf of all co-authors

**Responses:**

1. The periodicity analysis revealed cycles similar to the cycles of sunspot or ENSO activity. To support such ideal, it's better to provide more evidences. The periodicity analysis along is far from enough.

*The authors' response*: Comment accepted. According to your comments, the section 3.5 was removed from the revised paper because of the non-significant period of solar or ENSO and less contribution to this paper.

2. The Little Ice Age is generally abbreviated as LIA, instead of LAI or LA. Such expression should keep consistent in the MS (LA in line 1 of page 11).

*The authors' response*: Comment accepted. All such expressions "The Little Ice Age" in the MS were abbreviated as LIA.

3. Since the acceptable tree-ring chronology begins from 1660, it shouldn't be a 414-year temperature reconstruction (in the MS title), and "Results and Discussions" should based on the credible period (1660-2015).

***The authors' response***: Thank you for this suggestion. A reliable tree-ring chronology spanning 1660-2015 was developed on the basis of an EPS value greater than 0.85 (eleven trees). However, although an EPS value from AD 1600 to 1659 was less than 0.85, it matches a minimum sample depth of 6 trees in this segment. It is very important to extend the reconstruction tree-ring chronology as possible as we could because of few long climate reconstructions in this area. Moreover, the northern Hemisphere temperature series (D'Arrigo et al., 2006) and historical documents also partly confirmed that the reconstruction temperature from 1600 to 1659 was valuable. Therefore, we kept the reconstruction in this part. To help the readers to better understand this problem, we added some explanations (or illustration) of EPS in the segment of the chronology from 1600 to 1659 in the main text.

---

## Author Response (AR3)

**Dear Joel,**

Thank you very much for your helpful comments on our manuscript. We accepted all your comments and made correction carefully. All detailed revision and response are as below. Thank you again for all your help in processing our manuscript.

Sincerely yours,

Shanna Lyu and Xiaochun Wang,
on behalf of all co-authors

**Responses:**

1. Page 4, line 4: you do not use ARS after, then remove it.

*The authors' response*: Comment accepted. According to your comments, the ARS chronology was removed from the revised paper.

2. Page 8, line 3: Specify clearly that you will reconstruct MMT of April-July using the STD chronology only (because it is not clear that X in transfer function is the STD chronology.

*The authors' response*: Comment accepted. The sentence was modified as follow: where $Y$ was the April-July MMT and $X$ was the tree-ring index of the Korean pine STD chronology at year $t$.

3. Remove Fig.8 as it is not called in the text.

*The authors' response*: Comment accepted. According to your comments, the Fig.8 was removed from the revised paper.